# Significant Wave Height Retrieval Using XGBoost from Polarimetric Gaofen-3 SAR and Feature Importance Analysis

Tianran Song [1], Qiushuang Yan [1,*], Chenqing Fan [2,3], Junmin Meng [2,3], Yuqi Wu [1] and Jie Zhang [1,2,3]

1 College of Oceanography and Space Informatics, China University of Petroleum, Qingdao 266580, China
2 First Institute of Oceanography, Ministry of Natural Resources, Qingdao 266061, China
3 Technology Innovation Center for Ocean Telemetry, Ministry of Natural Resources, Qingdao 266061, China
* Correspondence: yanqiushuang@upc.edu.cn

**Abstract:** Empirical algorithms have become the mainstream of significant wave height (SWH) retrieval from synthetic aperture radar (SAR). But the plentiful features from multi-polarizations make the selection of input for the empirical model a problem. Therefore, the XGBoost models are developed and evaluated for SWH retrieval from polarimetric Gaofen-3 wave mode imagettes using the SAR features of different polarization combinations, and then the importance of each feature on the models is further discussed. The results show that the reliability of SWH retrieval models is independently confirmed based on the collocations of the SAR-buoy and SAR-altimeter. Moreover, the combined-polarization models achieve better performance than single-polarizations. In addition, the importance of different features to the different polarization models for SWH inversion is not the same. For example, the normalized radar cross section (NRCS), cutoff wavelength ($\lambda_c$), and incident angle ($\theta$) have more decisive contributions to the models than other features, while peak wavelength ($\lambda_p$) and the peak direction ($\varphi$) have almost no contribution. Besides, NRCS of cross-polarization has a more substantial effect, and the $\lambda_c$ of hybrid polarization has a stronger one than other polarization models.

**Keywords:** SWH retrieval; XGBoost; Gaofen-3 SAR wave mode; polarization; feature importance analysis

## 1. Introduction

Ocean waves form a complex random surface field that constantly changes under the influence of local winds and other environmental factors. Ocean waves can be described by wave spectrum or some statistical parameters. Significant wave height (SWH) is one of the most important statistical parameters of ocean waves, which is defined as the mean value of the maximum one-third of wave height. As well, it is approximately equivalent to the four times the square root of the integral of the measured wave spectrum in practical definition [1]. Spaceborne synthetic aperture radar (SAR) has been developed into the most powerful instrument for observing SWH from space at a fine spatial scale under all weather conditions [2]. In the traditional theoretical algorithms, the ocean wave spectrum is first retrieved from the SAR and then the SWH is computed via spectral integration. But these algorithms are complicated and have limited accuracies as the complex nonlinear distortion induced by the radial wave motions can cause image smearing and a loss of spectral energy [3]. Following the Seasat in 1978, dozens of satellites carrying SAR sensors have been launched and put into operation, providing tens of thousands of accessible SAR images. This promotes the development and mainstreaming of the data-driven empirical SWH inversion method.

Several empirical models have been developed to directly retrieve SWH from SAR images without explicitly retrieving the 2-D ocean wave spectrum. At the earliest, the CWAVE algorithm was proposed to retrieve SWH from C-band SAR data, such as ERS-2 SAR [4], Envisat ASAR [5], Sentinel-1 SAR [6], and RADARSAT-2 SAR [1]. The CWAVE model is

established based on linear regression or neural network, using 22 SAR features, including the normalized radar cross section (NRCS), the image variance (*cvar*), and 20 orthogonal components of the image spectrum (i.e., CWAVE spectral parameters, denoted as $S_1, \dots, S_{20}$). Besides, the strong correlation between SWH and azimuth cutoff wavelength ($\lambda_c$) in all sea states has been proved [7,8], and the linearly empirical dependency of the SWH on $\lambda_c$ has been established [9–11]. In the $\lambda_c$-based models, the peak wavelength ($\lambda_p$) and peak wave direction ($\varphi$) are often taken into account [9–12]. In addition to the features mentioned above, Stopa et al. [6] also considered incidence angle ($\theta$), image skewness (*skew*), and image kurtosis (*kurt*) in the neural network model for SWH retrieval. Currently, machine learning and deep learning have become mainstream methods for SAR SWH estimation because they are able to consider various SAR features and approximate nonlinear behavior without prior knowledge of the interrelationships between features [12–17].

In the early research, only the single-polarization (mostly vertical-vertical, VV) SAR features were exploited. The effects of polarized information on the empirical retrieval of SWH have been noticed recently. For example, Ren et al. established $\lambda_c$-based empirical models under different polarizations from RADARSAT-2 fine quad-polarization (HH (horizontal-horizontal), HV (horizontal-vertical), VH (vertical-horizontal), and VV) SAR images [9]. Collins et al. investigated the effect of polarization on the performance of the CWAVE models for SWH retrieval, by comparing the accuracy of SWH estimates at different polarizations of HH, HV, VH, VV, RH (right-circular-horizontal), RV (right-circular-vertical), RR (right-circular–right-circular), and RL (right-circular–left-circular) [1]. Wang et al. [18] explored the influence of introducing VH NRCS into the quadratic model constructed by VV features, and found that its introduction improved the performance of the model in high sea state. Then, the combinations of different polarizations have been considered to achieve higher performance of the SWH retrieval [19,20]. However, plentiful features will be considered if various polarizations are used. But the selection criteria of polarizations and SAR features are not clear when determining the model input parameter set, and it is unknown how each feature affects the performance of SWH inversion model.

Stepwise regression [21] and elastic net [22] methods have been applied to select significant terms in polynomial SWH inversion models, but only for the CWAVE features under VV polarization [1,4]. It is challenging for most machine learning models to determine the optimal feature set from plenty of polarized SAR features. As an improved algorithm based on the gradient boosting decision tree, Extreme Gradient Boosting (XGBoost) [23] is capable of obtaining the feature scores intelligently to reveal the importance of each feature on the training model [24], which provides experimental conditions for SAR feature contribution analysis. Thus, in this paper, we develop and validate the XGBoost models for SWH retrieval under different polarization combinations from Gaofen-3 SAR wave mode data, and we further discuss the importance of each SAR feature to different polarization models. The paper is organized as follows. We describe materials and methods in Section 2. Section 3 introduces establishment and validation of the XGBoost models under different polarizations. And Section 4 discusses the importance of each feature. The main conclusions are given in Section 5.

## 2. Materials and Methods

### 2.1. Gaofen-3 SAR Data

The Chinese Gaofen-3 satellite carrying a C-band SAR sensor has been in orbit since August 2016. Gaofen-3 SAR can operate in 12 imaging modes, of which wave mode is dedicated to ocean wave detection. In wave mode, Gaofen-3 SAR collects small SAR images (called imagettes) with an approximate coverage of 5 km × 5 km every 50 km along the flight direction with a nominal spatial resolution of 4 m over the open ocean. It provides quad-polarimetric (HH + HV + VH + VV) capability. In this paper, the Level-1A single-look complex (SLC) wave mode imagettes from 2016 to 2020 are collected. The SAR scenes contaminated by non-wave phenomena are rejected based on the following procedure: (1) The power-saturated data are rejected by checking the 'echoSaturation' value

given in the Gaofen-3 SAR product annotation file; (2) The imagettes contaminated by ice and land/island are excluded; (3) The homogeneity is checked according to the method proposed in Schulz-Stellenfleth [25]. The rejection percentage by the quality controls is approximately 30%, and finally, approximately 11,200 Gaofen-3 SAR imagettes are selected in this study (Figure 1).

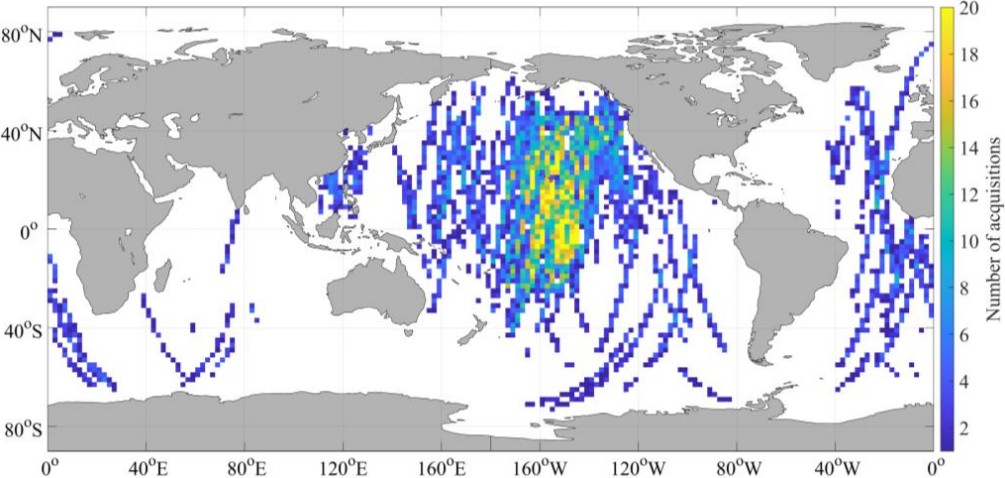

**Figure 1.** Map of the Gaofen-3 wave mode acquisitions for the year of 2016 to 2020 in data density for $2° \times 2°$ bins.

The maximum sensitivities to wave slopes have been noticed when using linear polarizations, and the polarimetric orientation modulation effects on ocean waves have been exploited [19,26–28]. Moreover, the possibility of ocean wave parameters inversion from compact polarization images has been demonstrated with RADARSAT data [1,29]. Therefore, in addition to conventionally processed SAR images at HH, HV, VH, and VV provided by Gaofen-3 wave mode, 45° linear polarization with a polarization orientation angle of 45° and compact polarizations known as RH, RV, RR, and RL are explored in this paper. According to Equations (3)–(6) in Collins et al. [1] and Equation (8) in Zhang et al. [28], we simulate the images at 45° linear polarization and compact polarizations from quad-polarization Gaofen-3 wave mode imagettes. In order to fully capture the nonlinearity of the imaging mechanism and the ocean wave response, we extract 28 SAR features for each polarization mentioned above.

The normalized radar cross section (NRCS) of SAR imagettes is directly related to ocean winds and thus can represent wave energy information of short wave roughness [19]. The Gaofen-3 NRCS values can be obtained from the following formula:

$$\sigma^0 = 10\log_{10}\langle DN \rangle - K \tag{1}$$

in which $\sigma^0$ is the NRCS in dB, *<DN>* denotes the mean value, $DN = I_s \times (qv/32767)^2$ denotes the image intensity, $I_s = I^2 + Q^2$ with $I$ ($Q$) being the value of real (imaginary) channel for the single-look complex SAR image, $qv$ is the maximum qualified value stored in the product annotation file according to the polarizations, $K$ is the calibration constant also stored in the product annotation file according to the polarization. However, the radiometric calibration of Gaofen-3 was proved to be inaccurate by comparing Gaofen-3 SAR images against Radarsat-2 and Sentinel-1 SAR data over land [30]. Only a tiny portion of the official Gaofen-3 wave mode products provide the $K$ values. Therefore, we apply the numerical weather prediction (NWP)-based ocean calibration method developed by Wang et al. [31] to estimate the values of $K$. The effectiveness of recalibration on SWH estimation using polarimetric Gaofen-3 wave mode SAR imagettes has been proved, and the recalibration constant values have been given in [20]. Figure 2 displays a typical example of all nine polarization NRCSs of Gaofen-3 imagettes, which is acquired on 7

February 2017, at 18:17 UTC. The images shown in Figure 2 are normalized by the min-max method, from which a clear wavy structure can be seen.

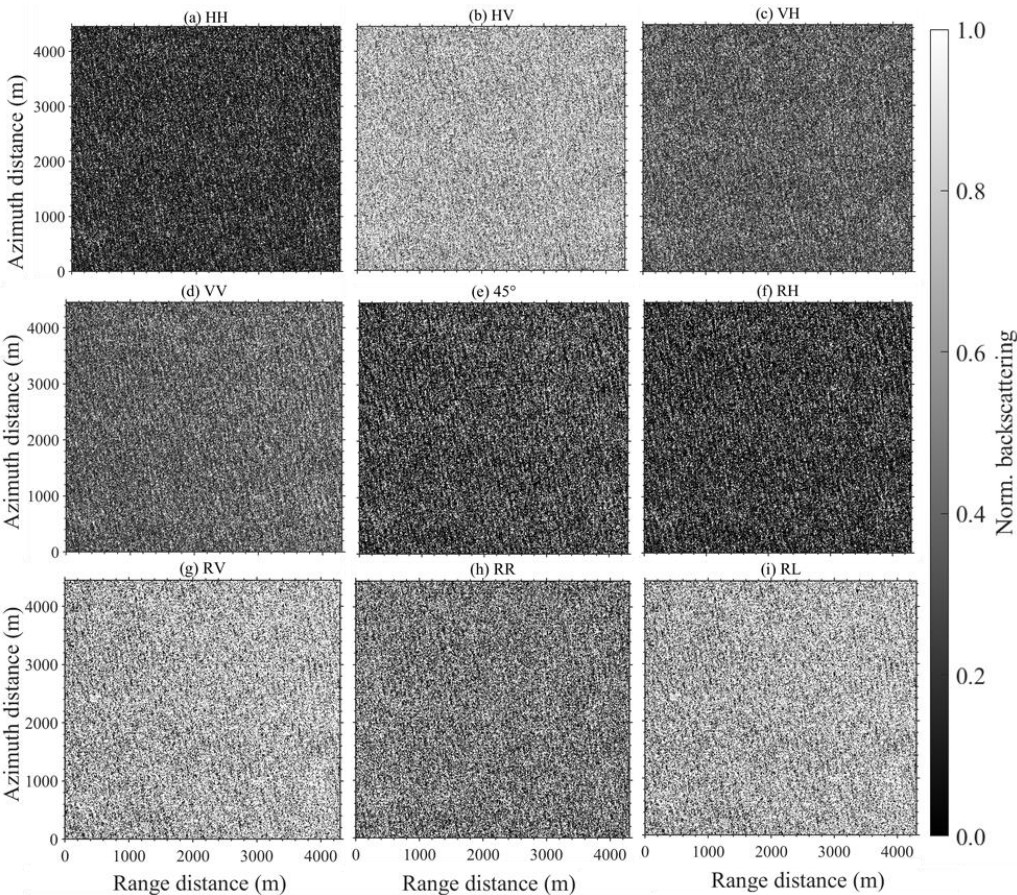

**Figure 2.** Case of Gaofen-3 SAR wave mode imagette acquired on 7 February 2017, at UTC 18:17:16. Image of normalized backscattering for (**a**) HH, (**b**) HV, (**c**) VH, (**d**) VV, (**e**) 45° linearly, (**f**) RH, (**g**) RV, (**h**) RR and (**i**) RL polarizations.

The normalized image variance (*cvar*) contains information on the sea state of longer waves, which is commonly used for SWH retrieval in empirical algorithms. It is defined as the variance of the Gaofen-3 image normalized by the mean intensity:

$$cvar = var\left(\frac{DN - \langle DN \rangle}{\langle DN \rangle}\right) \tag{2}$$

Besides, the higher-order features are considered by skewness (*skew*) and kurtosis (*kurt*) of the radar cross-section in this study:

$$skew = \frac{1}{n}\sum_{i=1}^{n}\left(\sigma_i^0 - \overline{\sigma^0}\right)^3 / s^3 \tag{3}$$

$$kurt = \frac{1}{n}\sum_{i=1}^{n}\left(\sigma_i^0 - \overline{\sigma^0}\right)^4 / s^4 \tag{4}$$

where $s$ is the standard deviation of $\sigma^0$.

Over the ocean, according to the SAR-ocean imaging mechanism of velocity bunching, the surface wave motions may distort the phase history of the backscattered signal leading to a nonlinear transformation between the local wave and the SAR image. As a result, the small wave components propagating near the azimuth direction may be blurred, which leads to a cutoff value. And the waves with wavelengths below the cutoff cannot be

resolved by SAR. Theoretically, the azimuth cutoff wavelength ($\lambda_c$) is determined by the orbital velocity of ocean waves over the integration time along with the range-to-velocity ratio ($\beta$) of the SAR platform. Hence, we choose $\lambda_c$ normalized by $\beta$ as another important feature. The $\lambda_c$ can be estimated by fitting a Gaussian function to the inter-correlation of the SAR cross-spectrum (real part) [32]. The Gaussian fit function is stated as follows:

$$C(x) = \exp(-(\frac{\beta x}{c})^2) \tag{5}$$

where $x$ denotes the spatial distance in the azimuth direction. However, the distortion of ocean waves also varies with respect to the peak wavelength ($\lambda_p$) and the peak direction ($\varphi$, defined relative to the range direction of the satellite) of the wave system [6]. Therefore, $\lambda_p$ and $\varphi$, which are computed from the cross-spectrum of SAR imagettes, are considered in our analysis.

It is proved that the introduction of the 20 CWAVE spectral parameters (denoted as $S_1, \ldots, S_{20}$) in empirical models leads to an even better performance of SWH retrieval. The CWAVE spectral parameters are obtained by integrating the orthogonal spectral components, which are calculated by mapping the SAR image modulation spectrum onto an orthogonal basis set of 20-nondimensional parameters. The orthogonal basis set is described in detail in the appendix of Schulz-Stellenfleth et al. [4]. In this paper, the CWAVE spectral parameters extracted from different polarimetric SAR imagettes are also applied in the XGBoost model for SWH inversion. Taking the VV polarization imagette in Figure 2 as an example, the 20 orthogonal spectral components are shown in Figure 3.

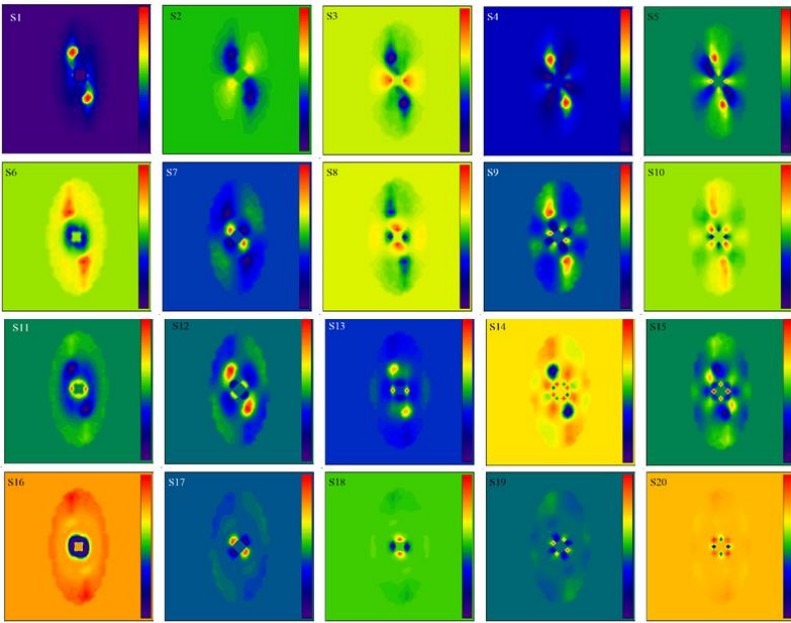

**Figure 3.** 20 orthogonal spectral components of the VV case in Figure 2d.

In addition, the incidence angle ($\theta$) of the Gaofen-3 wave mode could be switched from $20°$ to $50°$. It is an important feature that should be considered in building XGBoost models for SAR SWH retrieval.

### 2.2. Reference SWH

To maximize the number of colocations, we use SWH data from the fifth-generation ECMWF Atmospheric Reanalysis (ERA5) [33]. ERA5 data incorporate as many observations as possible into model estimates using advanced modeling techniques and state-of-the-art data assimilation systems, and provide new best estimates of the state of the atmosphere, ocean waves, and land surfaces. This dataset contains a number of ocean-wave variables

at a regular lat-lon grid of 0.5 degrees hourly, in which the significant height of combined wind waves and swell, i.e., SWH, is focused here. The consistency between ERA5 SWH and buoy SWH observations was validated in our previous study [20]. Each Gaofen-3 imagette is collocated with the time/space interpolated SWH from ERA5 yielding approximately 11,200 matched-up cells, and the collocated ERA5 SWHs roughly range from 0.3 to 8 m. All matchups are randomly grouped into three sets: training (60%), validation (20%), and test (20%). Both training and validation data are used to develop the models. The training set tunes the parameters (weights and biases) of the model while the validation set cross-validates and determines the hyperparameters. The importance analysis of Gaofen-3 SAR features for SWH estimate in different polarizations is also discussed based on the SAR-ERA5 data.

In addition, the SWH observations from buoys and altimeter are collected for independent verification in this study. Standard meteorological data of the 233 moored buoys in the waters operated by the National Data Buoy Center (NDBC) are collected. All the buoys are located more than 50 km from land and over 150 m deep. The quality of the NDBC SWH observations is very high, with an accuracy of approximately 0.2 m [34]. The SWH observations from Saral/AltiKa altimetry mission are selected as an additional data source for independent verification. The AltiKa radar altimeter was launched in 2013, on the Saral mission, a cooperative project between the Indian Space Research Organization (ISRO) and the French space agency Centre National d'Etudes Spatiale (CNES). AltiKa is operated at Ka-band, leading to several improvements in the performances for significant wave height (SWH) measurements [35]. The Gaofen-3 SAR imagettes are collocated respectively with the NDBC buoy and Saral altimeter SWH observations using the criteria of time separation within 1 h and spatial separation less than 100 km, yielding the matchups shown in Figure 4. This procedure yields only 43 SAR-buoy matching points, of which the buoy SWHs are mainly distributed in 2–3 m (Figure 5a). The collocation with Saral yields 757 points, of which the Saral SWHs are between 0.4–8 m (Figure 5b). To ensure an independent verification, the SAR-buoy and SAR-altimeter data are never seen by the models when tuning.

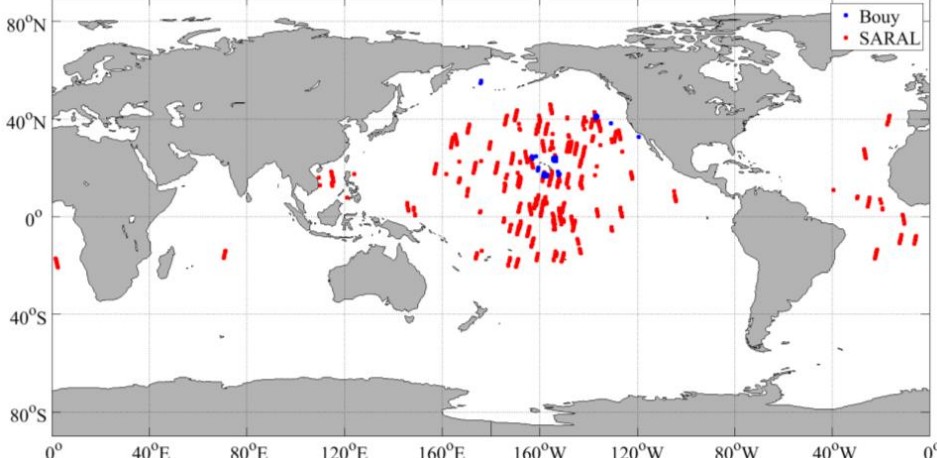

**Figure 4.** Map of collocations between Gaofen-3 wave mode data and SWH observations from NDBC buoys (blue) and Saral altimeter (red).

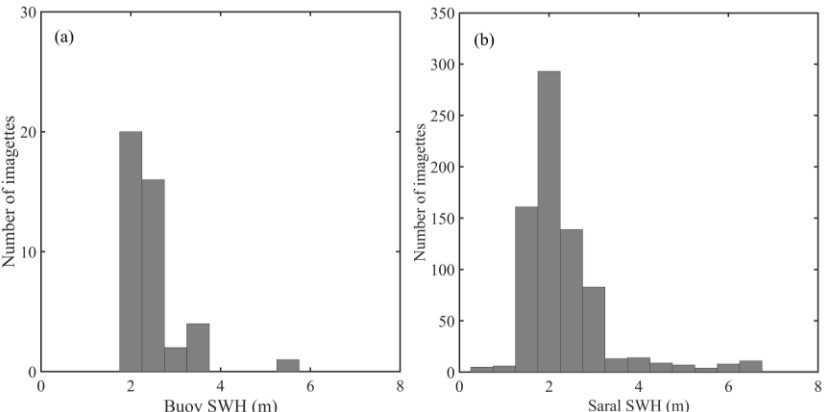

**Figure 5.** Histograms of the bouy SWH (**a**) and Saral SWH (**b**). The bin of the histograms is 0.4 m.

### 2.3. Extreme Gradient Boosting (XGBoost) Model

The XGBoost is an improved algorithm based on gradient-boosting decision trees. It can construct boosted trees efficiently and operate in parallel. Thus, the XGBoost is adopted in this study for the polarimetric Gaofen-3 SAR SWH retrieval. And the XGBoost model can be expressed as:

$$\hat{y}_i = \sum_{k=1}^{K} f_k(x_i), \; f_k \in F \; (i = 1, 2, \ldots n) \tag{6}$$

where $F$ is the set of decision trees, and n is the number of datasets. The core of the algorithm is to find the optimal parameters according to the principle of minimizing the objective function. The objective function can be written as:

$$Obj = L + \Omega \tag{7}$$

where, $L$ is a differentiable convex loss function that measures the difference between the prediction $\hat{y}_i$ and the target $y_i$: $L = \sum_{i=1}^{n} (y_i - \hat{y}_i)^2$. The model complexity function term $\Omega$ penalizes the complexity of the regression model. The additional regularization term helps to smooth the final learned weights to avoid over-fitting. In XGBoost, the second-order Taylor expansion of the objective function is carried out to find the parameters that minimize the objective function quickly. Therefore, the model complexity is well controlled while the model accuracy is ensured.

XGBoost obtains importance scores of features intelligently by the boosted tree construction, and the features that are used the most when boosting trees make key decisions and have the highest score [36]. The algorithm calculates the importance by "gain", "weight", or "cover" [37]. "Gain" refers to the average value of information gain optimization brought by a feature when the node is split. "Weight" is the number of times a feature is used to split the data across all trees. The "cover" is the relative value of a feature observation. In this study, the feature importance is set by "gain". According to Breiman et al. [38], the features that can significantly improve the estimated squared error are selected at each leaf node for a decision tree. For a particular feature, its importance is the sum of such squared improved performance over all nodes. The importance of a feature depends on whether the prediction performance changes considerably when such a feature is replaced with random noise. In addition, the hyperparameters of the XGBoost model are determined by minimizing the RMSE of SWH derived from Gaofen-3 SAR compared with ERA5 SWH with respect to the validation dataset, and the optimal hyperparameters for XGBoost in this paper are listed in Table 1.

**Table 1.** Optimal hyperparameter for XGBoost in this paper.

| Hyperparameter | Value |
|---|---|
| Number of estimators | 200 |
| max_depth | 50 |
| learning_rate | 0.05 |
| reg_lambda | 1 |
| reg_alpha | 0 |
| min_child_weight | 1 |
| gamma | 0 |
| subsample | 1 |

## 3. Results

Wang et al. [19] compared SWH derived from single-, dual- and quad-polarized deep convolutional neural networks, and found that exploitation of SAR quad-polarimetry (HH + HV + VH + VV) information would improve SAR wave height retrievals under high sea conditions. Inspired by this, in order to utilize the polarimetry information provided by Gaofen-3 wave mode imagettes, we also consider six combined-polarization modes: HH + HV, VV + VH, RH + RV, RL + HV, Quad (HH + HV + VH + VV) and All (HH + HV + VH + VV + 45° + RH + RV + RR + RL) in addition to the nine single-polarization modes mentioned in Section 2.1. The XGBoost models under 15 polarization modes are established based on the collocated data set of 12,000 Gaofen-3 imagettes matched with SWH from ERA5 in this section. The prediction accuracies of SWH are independently assessed by the SWH observations from the NDBC buoys and Saral/AltiKa altimetry mission.

Figure 6 shows the color-coded correlation coefficient matrices for the features of different polarization XGBoost models. We found that many of the features are highly correlated, especially for the combined-polarization models. Therefore, we employ recursive feature elimination (RFE) [39] to clarify the role of each feature on the SWH retrieval and to remove redundant ones effectively. Based on the "feature_importances" toolkit of XGBoost, the feature that ranks the last in importance is eliminated in each iteration, and the remaining features will create subsets to build the SWH estimation model again [40]. The XGBoost regression model is retrained with each subset, and the model performance is also calculated with RMSE of the inversion SWH compared with ERA5 SWH from test datasets.

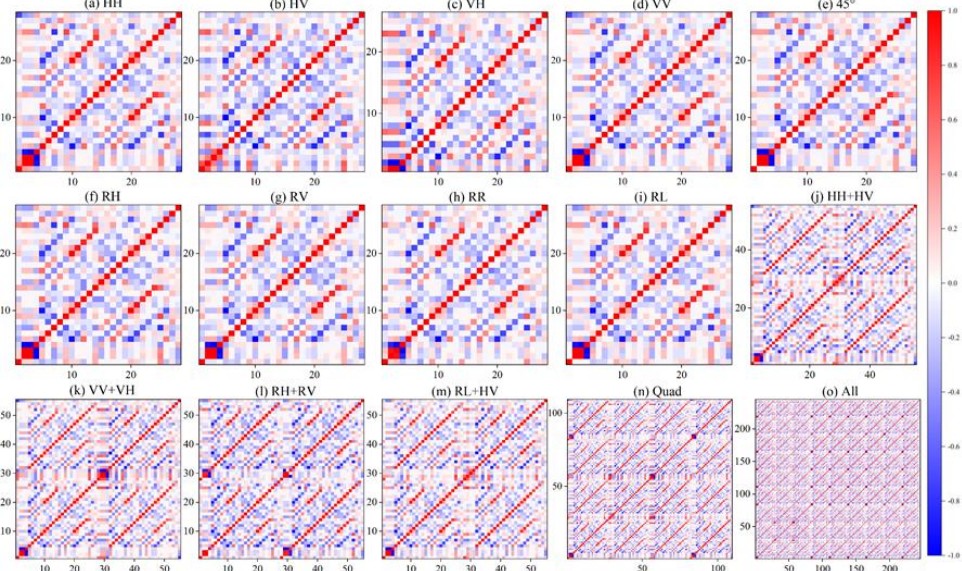

**Figure 6.** Color-coded correlation coefficient matrices for the features in different polarization models of (**a**) HH, (**b**) HV, (**c**) VV, (**d**) VH, (**e**) 45° linearly, (**f**) RH, (**g**) RV, (**h**) RR, (**i**) RL, (**j**) HH + HV, (**k**) VV + VH, (**l**) RH + RV, (**m**) RL + HV, (**n**) Quad, (**o**) All.

The corresponding relationship between the number of remaining features and the performance of 15 XGBoost models are given in different colors in Figure 7a. We normalize the number of remaining features of all the models to the range of 0–1 to facilitate the comparison between different polarizations. With the same fraction of remaining features, there is little difference in the performances of the models under nine single-polarization modes (HH, HV, VH, VV, 45° linearly, RH, RV, RR, and RL). The best performance appears in a 45° linearly polarized model with the lowest RMSE. The combined-polarizations have lower RMSEs than the single-polarizations (about 3 cm lower on average), except for RH + RV. This indicates that the increase of polarization information may improve performance, but it is not inevitable. For a specific polarization mode, the SWH estimation performance improves with the increasing of the number of remaining features. However, the improvement is not apparent and even oscillates when the fraction of remaining features exceeds 0.4, which means that many input features are redundant or even lead to worse model performance.

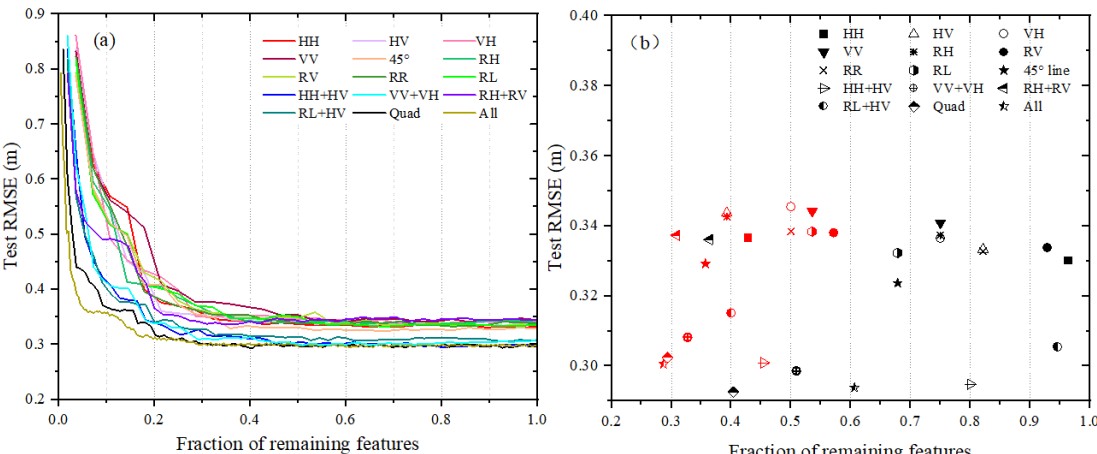

**Figure 7.** The performance of XGBoost models for SWH retrieval using different input features under 15 polarizations compared with ERA5 SWH. (**a**) The relationship of the fraction of remaining features and RMSEs of the SWH derived from XGBoost under 15 polarization modes, with different colors representing different polarizations. (**b**) Performance of XGBoost models based on the optimal feature sets (red) and the best feature sets (black) under 15 polarization modes (different symbols).

Therefore, the input features of each polarization model should be redetermined to remain the essential features and remove redundant features as possible. Here, we select features from high to low according to the importance ranking provided by XGBoost, so we only need to determine the required number of input features. For this, we propose two options: (1) Best number of feature set: the number of features that make the model performance achieve the best performance ($RMSE_{min}$); (2) Optimal number of feature set: the minimum number of features that is used when the model performance is not much worse than from the best performance ($RMSE \leq RMSE_{min} + 0.01$ m). Figure 7b shows the RMSEs of models using the best feature sets (black symbols) and the optimal feature sets (red symbols).

Obviously, for most polarization models, the fractions of the best feature sets are between 0.8 and 1, while the fractions of the optimal feature sets are below 0.5, indicating that for the optimal feature sets, we lose few accuracies but halve the features. The optimal features used in different polarization models are documented in Table A1. Moreover, the RMSEs of SWH derived from 15 polarization models using different input features, containing all the features extracted from SAR imagettes, the best feature sets, and the optimal feature sets are listed in Table 2. Among the single-polarization models constructed using the optimal feature sets, the 45° linear polarization model achieves the lowest RMSE (0.329 m) with the fewest features (only 10). While for the combined-polarizations, it is

clear that HH + HV and VV + VH exhibit excellent performance, achieving low RMSEs of 0.300 m and 0.308 m using the optimal feature sets (25 and 18 features, respectively). The VV + VH model performs better and uses fewer features than single-polarization models with the best feature sets. This suggests that with a comparable number of inputs, the combination of co-polarized and cross-polarized SAR features may perform better for SWH estimation than that using either co-polarized or cross-polarized ones alone. This is also true for RL + HV (co-polarization + cross-polarization) model, but its performance is not as good as that of the HH + HV and VV + VH models. The RH + RV model does not have such an advantage. In addition, the accuracies of the Quad and All polarization models are not much different from those of the HH + HV and VV + VH models, but more input features are used.

**Table 2.** The number of inputs and the performance of 15 polarization models for SWH retrieval using different input combinations, including all features extracted from SAR imagettes, the best feature sets, and the optimal feature sets.

| Polarization | All Features | | Best Feature Sets | | Optimal Feature Sets | |
|---|---|---|---|---|---|---|
| | Number | RMSE (m) | Number | RMSE (m) | Number | RMSE (m) |
| HH | 28 | 0.332 | 27 | 0.330 | 12 | 0.337 |
| HV | 28 | 0.339 | 23 | 0.333 | 11 | 0.343 |
| VH | 28 | 0.342 | 21 | 0.336 | 14 | 0.345 |
| VV | 28 | 0.344 | 21 | 0.340 | 15 | 0.344 |
| 45° | 28 | 0.328 | 19 | 0.323 | 10 | 0.329 |
| RH | 28 | 0.340 | 21 | 0.337 | 11 | 0.342 |
| RV | 28 | 0.335 | 26 | 0.334 | 16 | 0.338 |
| RR | 28 | 0.338 | 23 | 0.332 | 14 | 0.338 |
| RL | 28 | 0.336 | 19 | 0.332 | 15 | 0.338 |
| HH + HV | 55 | 0.301 | 44 | 0.295 | 25 | 0.300 |
| VV + VH | 55 | 0.309 | 28 | 0.299 | 18 | 0.308 |
| RH + RV | 55 | 0.344 | 20 | 0.336 | 17 | 0.337 |
| RL + HV | 55 | 0.307 | 52 | 0.306 | 22 | 0.315 |
| Quad | 109 | 0.297 | 44 | 0.293 | 32 | 0.302 |
| All | 244 | 0.297 | 148 | 0.294 | 58 | 0.301 |

One can see from Figures 8 and 9 that, for all the polarization modes, reasonably good SWH estimates could be achieved via the XGBoost model using optimal feature sets under moderate sea conditions (roughly 1–4 m), where the mean lines almost overlap the one to one straight line and the residuals are close to zero. This probably results from two reasons: (1) all polarizations work well at moderate sea states; (2) the data are primarily distributed in moderate seas. However, overestimation/underestimation could be found under low/high sea conditions. Among the single-polarizations, the cross-polarizations (HV, VH, and RR) show slightly smaller underestimation in high wave regime than the co-polarizations (HH, VV, and RL). However, the 45° linear polarization achieves the best estimate in both low and high sea conditions. In addition, under low sea conditions, the mean bias of HV/VH polarization is slightly larger than that of HH/VV polarization probably due to the low signal-to-noise ratio of the cross-polarization SAR images. But the variance of bias is larger under the co-polarizations, which is mainly caused by several scenes with poor quality under the co-polarizations (such as the anomalously overestimated points appearing in the low sea states shown in Figure 8d). The combined-polarization modes such as HH + HV, VV + VH, RL + HV, and Quad (HH + HV + VH + VV) and All (HH + HV + VH + VV + 45° + RH + RV + RR + RL) achieve better performance by combining the advantages of co-polarization and cross-polarization. RH + RV performs poorly at high sea states. And we can see that the Quad mode has a slight advantage over HH + HV or VV + VH at high sea conditions, but All polarization does not perform even better, indicating that the performance of SWH estimation saturates with the increasing of polarization information.

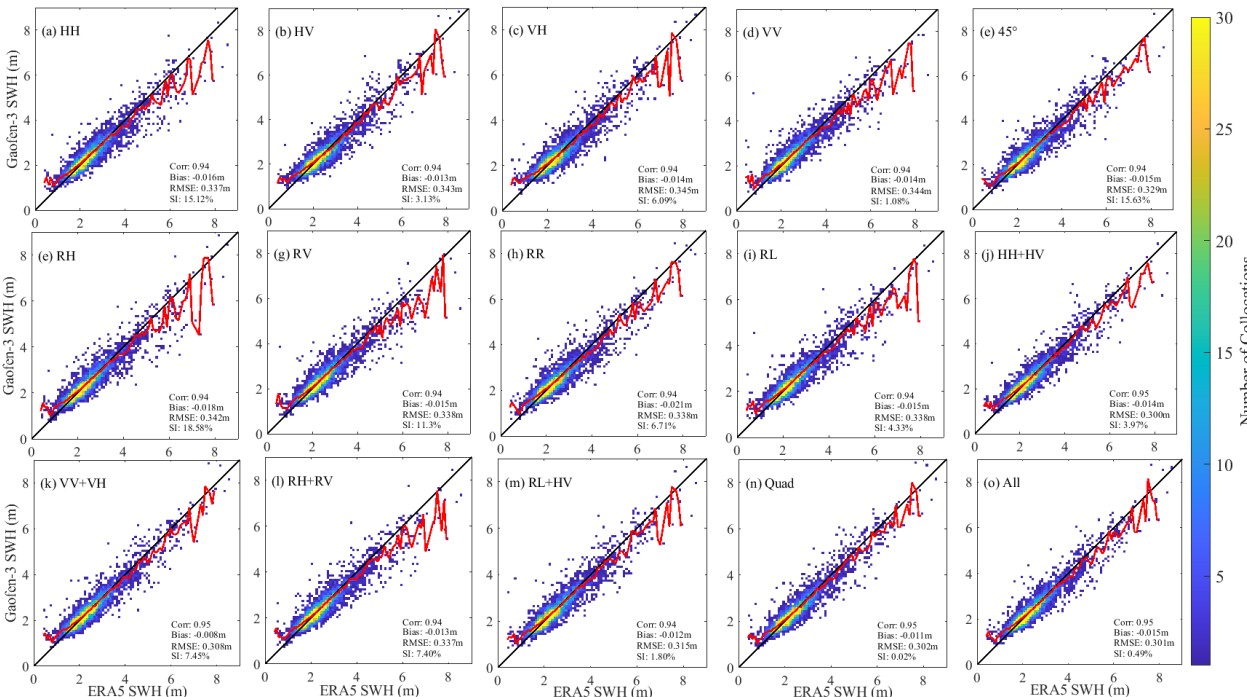

**Figure 8.** Plots of Gaofen-3 SWH retrievals from the XGBoost model using optimal feature sets versus ERA5 SWH for the 15 polarization modes of (**a**) HH, (**b**) HV, (**c**) VV, (**d**) VH, (**e**) 45° linearly, (**f**) RH, (**g**) RV, (**h**) RR, (**i**) RL, (**j**) HH + HV, (**k**) VV + VH, (**l**) RH + RV, (**m**) RL + HV, (**n**) Quad, (**o**) All. The solid red lines join the mean values from SAR estimates in each 0.1 m bin of ERA5 SWH. Colors denote the data numbers within 0.1 m × 0.1 m bins.

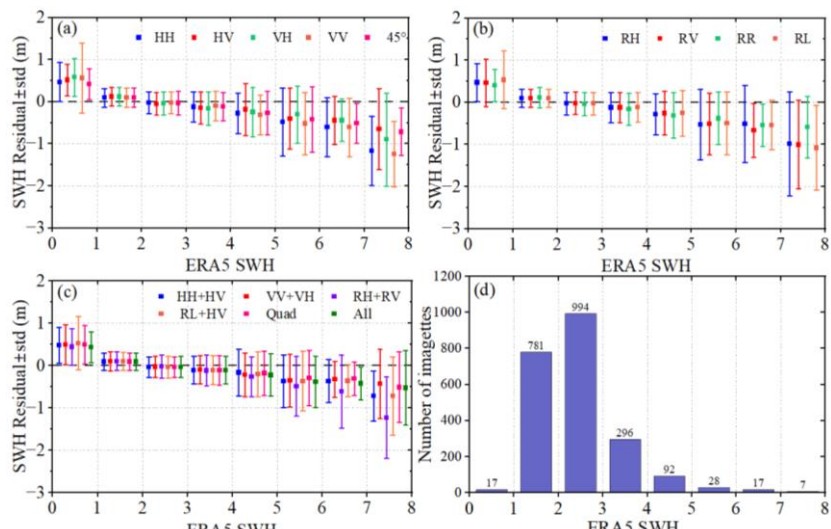

**Figure 9.** Comparison of SWH residuals against ERA5 SWH, with error bars presenting the standard deviation. The Gaofen-3 SAR SWH estimates were obtained from the XGBoost model under the 15 polarization modes of (**a**) HH, HV, VH, VV, 45°; (**b**) RH, RV, RR, RL; and (**c**) HH + HV, VV + VH, RH + RV, RL + HV, Quad, All. (**d**) Histogram of ERA5 SWH in bin size of 1 m, where the data count is labeled in black text.

Figure 10 presents the comparison of Gaofen-3 SWH from XGBoost models using optimal feature sets against the measurements from NDBC buoys and Saral altimeter for 15 polarization modes. Combined-polarizations have prominent advantages for SAR SWH estimation compared with the single-polarizations. And HH + HV polarization is still the best performer. For most polarization models, SAR-buoy comparison has a smaller RMSE

than SAR-altimeter because the buoys tend to be located in waters with more moderate sea states. Taking HH + HV as an example, we achieve a good agreement against the altimeter SWH, presenting 0.334 m RMSE, 0.95 correlation coefficient, and 1.94% SI, which is close to the current state-of-the-art model for SWH retrieval from Gaofen-3 wave mode with 0.32 m RMSE [19]. However, the latter uses a convolutional network deep learning model with two-dimensional cross-spectrum input, which has a more complex model structure and and lower efficiency. The RMSE of the worst-performing model is also less than 0.5 m, indicating that the XGBoost model is reliable when applied to Gaofen-3 wave mode SAR imagettes for SWH inversion.

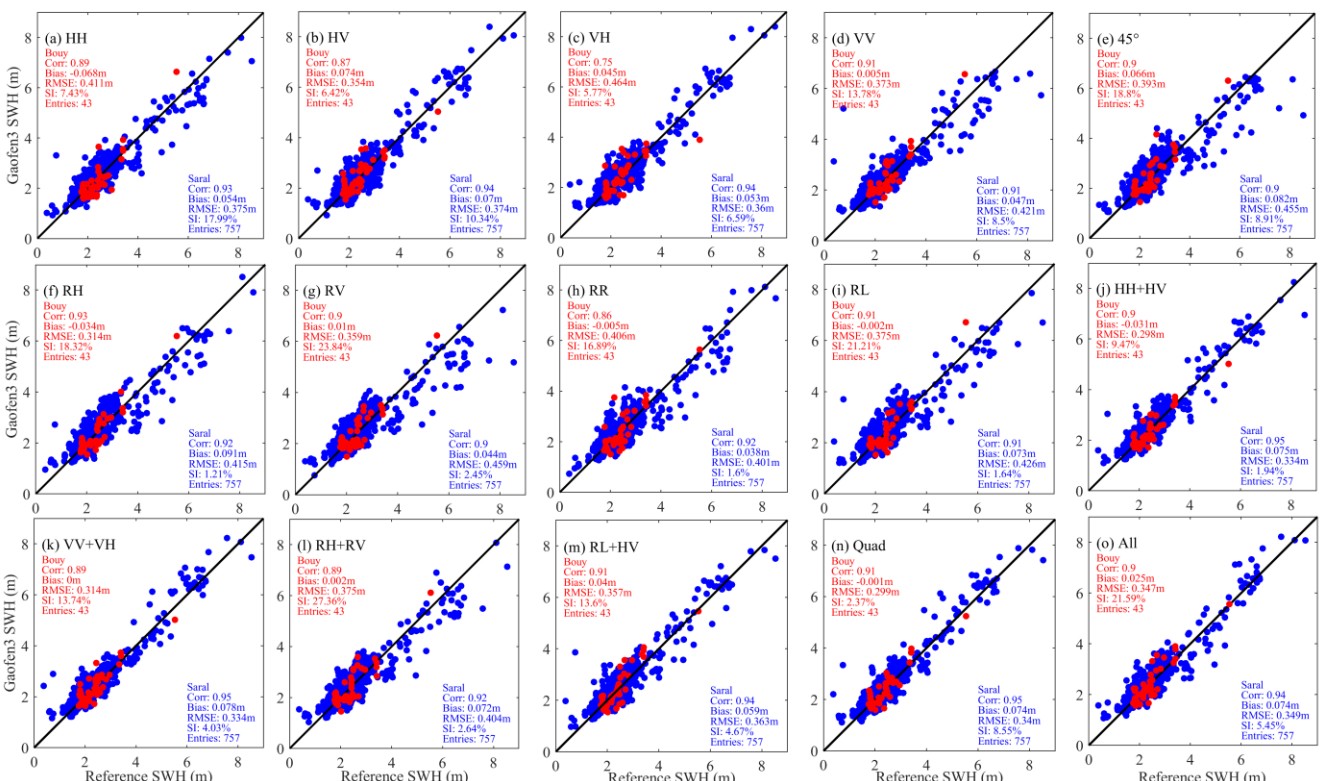

**Figure 10.** Plots of Gaofen-3 SWH retrievals from XGBoost models using optimal feature sets versus SWH measurements from buoys (red) and Saral (blue) for the 15 polarization modes of (**a**) HH, (**b**) HV, (**c**) VV, (**d**) VH, (**e**) 45° linearly, (**f**) RH, (**g**) RV, (**h**) RR, (**i**) RL, (**j**) HH + HV, (**k**) VV + VH, (**l**) RH + RV, (**m**) RL + HV, (**n**) Quad, (**o**) All.

## 4. Discussion

The construction of the de-redundant XGBoost model is implemented in Section 3, and it achieves good SWH estimation precision. However, the effects of each feature on SWH estimation in different polarization models are still unknown. Therefore, in this section, we carry out the importance analysis focusing on the extracted SAR features from Gaofen-3 SAR data.

### 4.1. The Importance of NRCS

The importance of NRCS to SWH retrieval using the XGBoost model under different polarization modes is explored. Figure 11a presents the importance ranking of NRCS provided by XGBoost models under single-polarizations. The ranking is normalized to a 0–1 range, and the feature with a smaller ranking value is more important. The dotted line in the figure represents the normalized numbers of optimal feature sets determined in Section 3. Besides, additional models are trained with the same architecture and hyperparameters but with NRCS removed to assess the dependence of performance of the models on NRCS. Here, relative RMSE, defined as the ratio of the change in RMSE due to the NRCS removal

to the RMSE using all the features, is adopted to represent its influence on the accuracy of the models. Obviously, the feature positively affects the SWH inversion if the relative RMSE is greater than 0, and the greater the absolute value, the greater the effect on the model accuracy. The relative RMSE values of NRCS for SWH inversion under the single-polarization models are shown in Figure 11b.

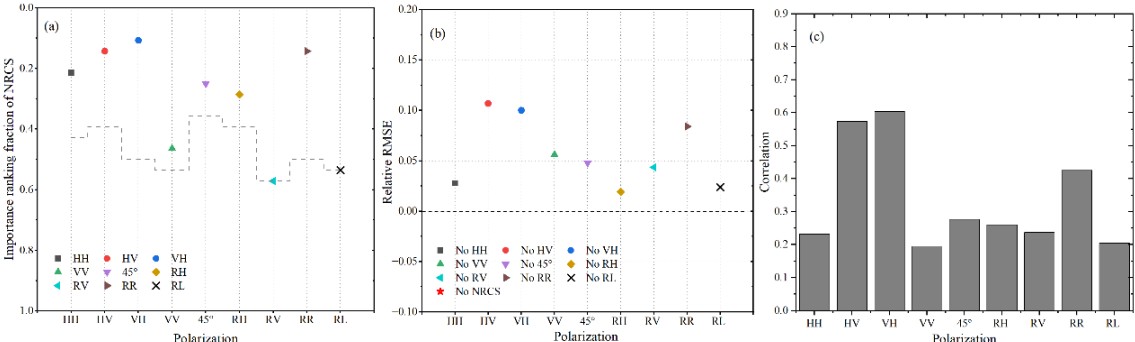

**Figure 11.** The importance of NRCS to SWH retrieval from single-polarization models. (**a**) The importance ranking of NRCS provided by XGBoost models normalized to 0–1. The dotted line represents the optimal feature sets under different polarizations. (**b**) The effect of removing NRCS on the SWH estimation from different polarized models. Different markers in various colors represent NRCSs of different polarizations. (**c**) The correlation between NRCS and ERA5 SWH under different polarizations.

In the single-polarization models, the NRCS obtains a high-importance ranking from XGBoost. As an important contribution term, it is selected into the optimal feature sets. NRCS shows a significant positive effect on model accuracy at all single-polarizations. In addition, the importance of NRCS in different polarizations is different. In the cross-polarization channels (HV, VH, and RR), the NRCSs have higher rankings and make more significant positive contributions to the model than in the co-polarizations (HH, VV, and RL) and hybrid-polarizations (45° linely, RH and RV). This is consistent with the correlations between NRCS and ERA5 SWH under different polarizations presented in Figure 11c.

The cases for the combined-polarizations are shown in Figure 12. There are at least two NRCSs of different polarizations in the input of each model, but not all NRCSs are necessary for SWH retrieval from the XGBoost model. Among the three dual-polarization models of HH + HV, VV + VH, and RL + HV, which combine co-polarization and cross-polarization, the cross-polarized NRCS ranks so high that it is selected by the optimal feature set and has a significant impact on the SWH inversion accuracy. In contrast, the co-polarized NRCS is not selected due to the low importance ranking, and it does have little impact on the accuracy. In the RH + RV model, the RH NRCS has a higher importance ranking but a negative effect on the model accuracy. However, with RH NRCS removed, the performance of the model using the optimal feature set deteriorates, indicating that there are redundant items that interact with RH NRCS resulting in a negative effect on the model. Moreover, for Quad (HH + HV + VH + VV) and All (HH + HV + VH + VV + 45° + RH + RV + RR + RL) polarizations, the rankings of cross-polarized NRCSs are still at the forefront so that they can be selected by the optimal feature sets. The cross-polarized NRCSs have more excellent effects on the accuracy of the SWH inversion than the one of other polarizations. However, we noticed that although HV NRCS has a high ranking in the Quad model, its removal improves the performance, which may be due to the negative effects of the nonlinearity between HV NRCS and other input features (such as $S_{2\_VV}$). On the other hand, for Quad or All polarization modes, removing NRCS of only one certain polarization has little effect on the model, mainly because there are too many inputs for the models (109 and 245, respectively). But the model performance drops significantly with all NRCSs removed, demonstrating that the mathematical interaction between NRCSs of different polarizations has an apparently positive effect on SWH estimation.

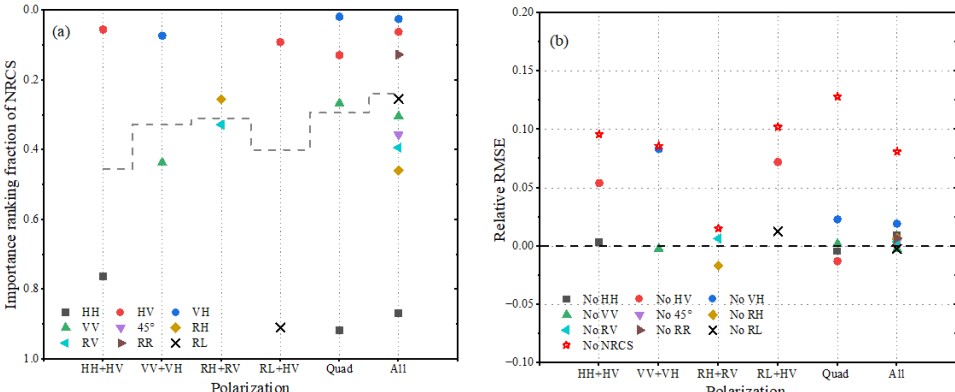

**Figure 12.** The importance of NRCS to SWH retrieval from combined-polarization models. (**a**) The importance ranking of NRCSs provided by XGBoost models, normalized to the range of 0–1. (**b**) The effect of removing NRCS on the SWH estimation from different polarized models. The red pentagrams represent the relative RMSE corresponding to the removal of NRCS-terms in the combined-polarization models.

To determine the effect of sea state on the importance of NRCS to SWH retrieval using the XGBoost model under different polarizations, the collected Gaofen-3 wave mode imagettes are divided into three groups according to the sea state: low (<1 m), medium (1–4 m), and high (>4 m). The relative RMSEs of the polarized models caused by removing NRCS-terms in different sea states are presented in Figure 13a. For the models containing cross-polarization information, the NRCS-terms have a great effect on their accuracies in high sea conditions, which is consistent with the strong correlation between cross-polarized NRCS and ERA5 SWH at high sea states, as shown in Figure 13b. But at low sea states, the cross-polarized NRCSs have lower correlations with SWH, probably due to the facts that the VH and HV products are more impacted by noise (the signal is too weak) at low wind. Although the co-polarized NRCSs have higher correlations at low sea states, they have little effects on the performance of the single-polarization models, which is mainly because of the several images with poor quality in co-polarization. For the Quad and All models, the effects of the NRCS-terms on the model accuracies are in the leading position at both low and high sea states, indicating that the combination of co-polarized and cross-polarized NRCSs improves the SWH inversion in extremely low and high sea conditions.

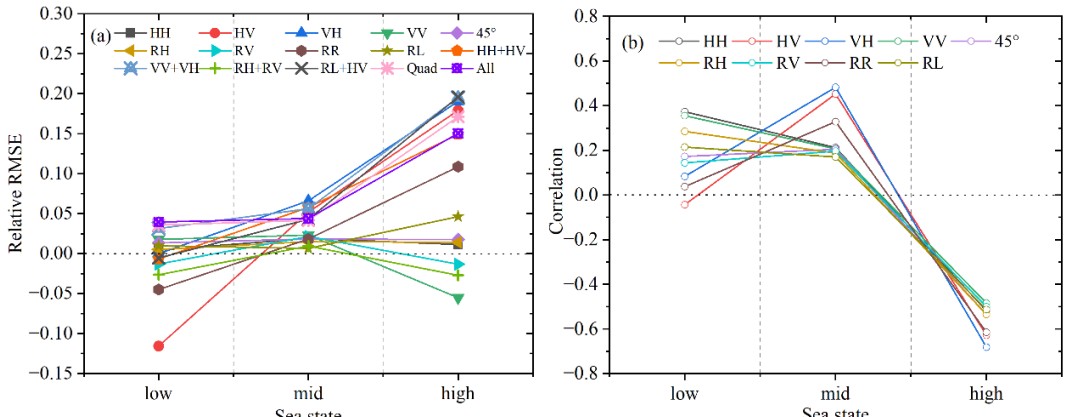

**Figure 13.** (**a**) The effects of removing NRCS-terms on the model performance of different polarizations under different sea states. (**b**) The correlation between NRCS of different polarizations and SWH under different sea states.

### 4.2. The Importance of Cvar

As shown in the Figure 14a, the ranking of *cvar* in single-polarization models is lower than that of NRCS. As well, the *cvar* is only remained by the optimal feature set in cross-polarization (HV, VH, and RR) models, while it is removed as redundancies in other models. However, the removal of *cvar* has little or even a negative effect on the accuracy of the SWH inversion from the cross-polarization models, as shown in Figure 14b. Paradoxically, the correlation between *cvar* and SWH, shown in Figure 14c, is not low. We believe there are terms in each polarization model that either act repetitively with *cvar* or have nonlinear effects with *cvar*, which may negatively affect the model. Taking HV polarization as an example (both the correlation and the importance ranking are high, but the contribution of *cvar* to the model is negative), $S_{14}$ has repeated effects with *cvar*. Moreover, the performance of the model is better when $S_1$ and *cvar* are removed than when both are included in input parameter set. And it becomes best when only $S_1$ or *cvar* is included, indicating that the nonlinearity between the two terms degrades the SWH estimation accuracy.

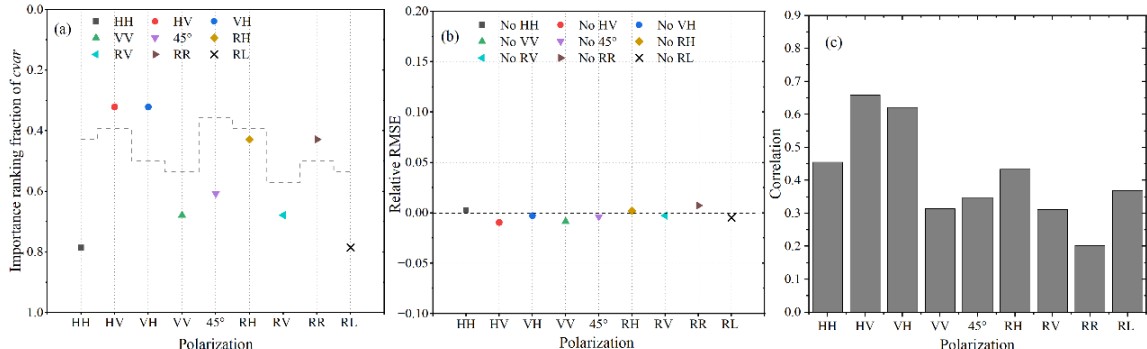

**Figure 14.** The importance of *cvar* to SWH retrieval from single-polarization models. (**a**) The importance ranking of *cvar* provided by XGBoost models normalized to 0–1. The dotted line represents the optimal feature sets under different polarizations. (**b**) The effect of removing *cvar* on the SWH estimation from different polarized models. Different markers in various colors represent *cvar* of different polarizations. (**c**) The correlation between *cvar* and ERA5 SWH under different polarizations.

From Figure 15a, it can be seen that the importance ranking of different polarized *cvar* is not the same in different combined-polarization models. In the models containing cross-polarization information (HH + HV, VV + VH, RL + HV, Quad, and All), the *cvar*s of cross-polarizations are all ranked high and are selected as the optimal features, while other polarized *cvar*s are mostly rejected. However, as shown in Figure 15b, even cross-polarized *cvar*s have little effect on SWH inversion performance. The performance of the model does not change much when removing all *cvar*-terms in the input parameters. The contradictions similar to the single-polarization models occur in the combined-polarizations. This is also probably because of the repetitive and nonlinear effects between *cvar* and other features.

### 4.3. The Importance of Skew and Kurt

Figures 16 and 17 demonstrate the importance of *skew* and *kurt* to the SWH inversion from single-polarization models. The importance ranking of *skew* provided by XGBoost is higher than that of *kurt* under a certain polarization. Only two models (HH and RR) do not select *skew* as the optimal feature parameter, while only two models (RV and RL) select *kurt*. Both *skew* and *kurt* have little effects on the inversion accuracy of SWH, with *kurt* playing a negative role in more models. Interestingly, the correlation of *skew* with SWH is almost opposite to that of *kurt* under every polarization, though the absolute value of the latter is slightly smaller. Actually, the correlations of both features with SWH are not small, but the complex mathematical interaction (repeatability or nonlinearity) between the features limits the importance of the two for the SWH inversion from different polarization models.

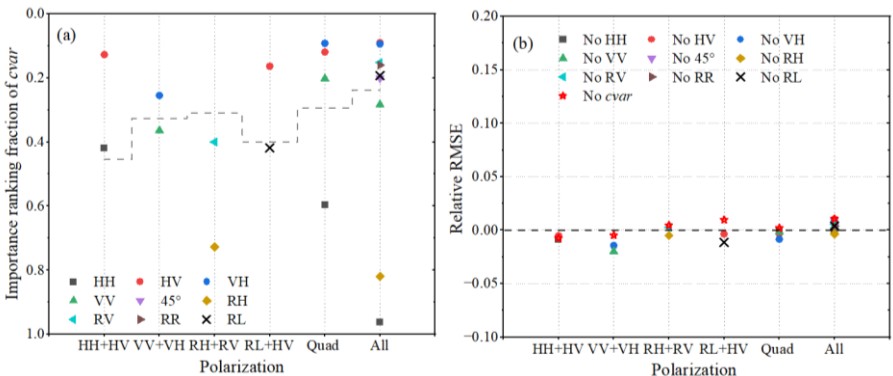

**Figure 15.** The importance of *cvar* to SWH retrieval from combined-polarization models. (**a**) The importance ranking of *cvar* provided by XGBoost models, normalized to the range of 0–1. (**b**) The effect of removing *cvar* on the SWH estimation from different polarized models. The red pentagrams represent the relative RMSE corresponding to the removal of *cvar*-terms in the combined-polarization models.

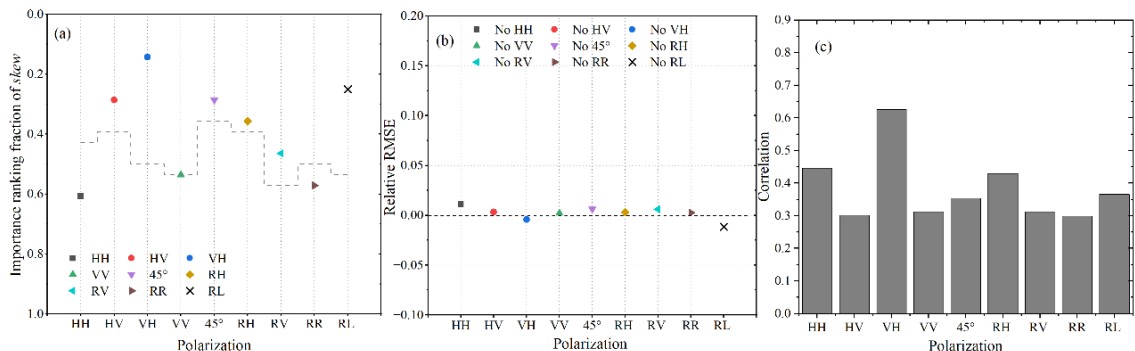

**Figure 16.** The importance of *skew* to SWH retrieval from single-polarization models. (**a**) The importance ranking of *skew* provided by XGBoost models normalized to 0–1. The dotted line represents the optimal feature sets under different polarizations. (**b**) The effect of removing *skew* on the SWH estimation from different polarized models. Different markers in various colors represent *skew* of different polarizations. (**c**) The correlation between *skew* and ERA5 SWH under different polarizations.

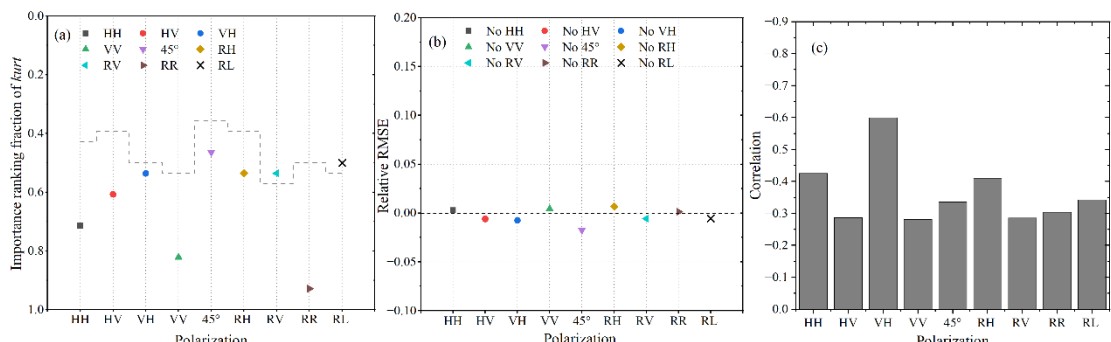

**Figure 17.** The importance of *kurt* to SWH retrieval from single-polarization models. (**a**) The importance ranking of *kurt* provided by XGBoost models normalized to 0–1. The dotted line represents the optimal feature sets under different polarizations. (**b**) The effect of removing *kurt* on the SWH estimation from different polarized models. Different markers in various colors represent *kurt* of different polarizations. (**c**) The correlation between *kurt* and ERA5 SWH under different polarizations.

For the combined-polarizations, as shown in Figures 18 and 19, the influence of the complex mathematical relationship between different features on the importance of *skew* and *kurt* to SWH inversion is more pronounced. In the HH + HV and VV + VH models, the optimal feature sets select all *skew*-terms and co-polarized *kurt*s. However, all the *skew*- and

*kurt*-terms negatively affect the SWH inversion accuracy. In the RH + RV model, only RV *skew* is selected to be a significant term, but the removal of RV *kurt* causes a more obvious impact on SWH retrieval. In the RL + HV model, all the *skew*- and *kurt*-terms are selected by the optimal feature set, but the effect of *skew* is smaller than that of *kurt*. And the model accuracy is better when both *kurt*-terms are removed though the HV *kurt* has a positive impact on the SWH estimation accuracy. This demonstrates a nonlinear effect between the RL- and HV-*kurt*, which reduces the accuracy of the SWH estimation. As for the Quad and All models, the rankings of *skew* and *kurt* are different, but neither has significant effects on SWH retrieval.

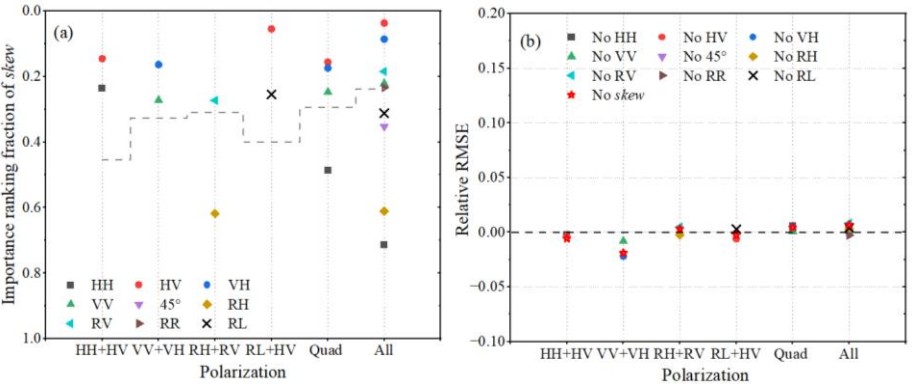

**Figure 18.** The importance of *skew* to SWH retrieval from combined-polarization models. (**a**) The importance ranking of *skew* provided by XGBoost models, normalized to the range of 0–1. (**b**) The effect of removing *skew* on the SWH estimation from different polarized models. The red pentagrams represent the relative RMSE corresponding to the removal of *skew*-terms in the combined-polarization models.

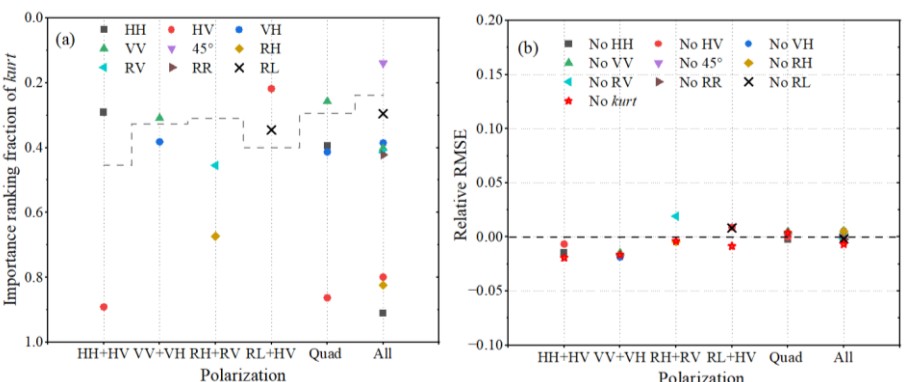

**Figure 19.** The importance of *kurt* to SWH retrieval from combined-polarization models. (**a**) The importance ranking of *kurt* provided by XGBoost models, normalized to the range of 0–1. (**b**) The effect of removing *kurt* on the SWH estimation from different polarized models. The red pentagrams represent the relative RMSE corresponding to the removal of *kurt*-terms in the combined-polarization models.

### 4.4. The Importance of $\lambda_c/\beta$

Figure 20 demonstrates the high importance of $\lambda_c/\beta$ to SWH estimation in single-polarizations. The $\lambda_c/\beta$ is selected as an important contribution term being one of the optimal features in each polarization due to the high ranking from XGBoost. The $\lambda_c/\beta$ is highly correlated with SWH and it significantly affects the inversion accuracy of most single-polarization models. However, the importance of the $\lambda_c/\beta$ varies in different polarization models. The $\lambda_c/\beta$ performs similarly weaker with lower correlation and lower importance for SWH estimation in the HV and VH polarization models. Especially for the VH model, it makes a negative effect on the model. The importance ranking of $\lambda_c/\beta$-term is extremely high in the polarizations except for HV and VH. But only in the 45° linearly, RV, and RL

polarizations, the positive effect of the cutoff wavelength on the SWH estimation reaches its maximum.

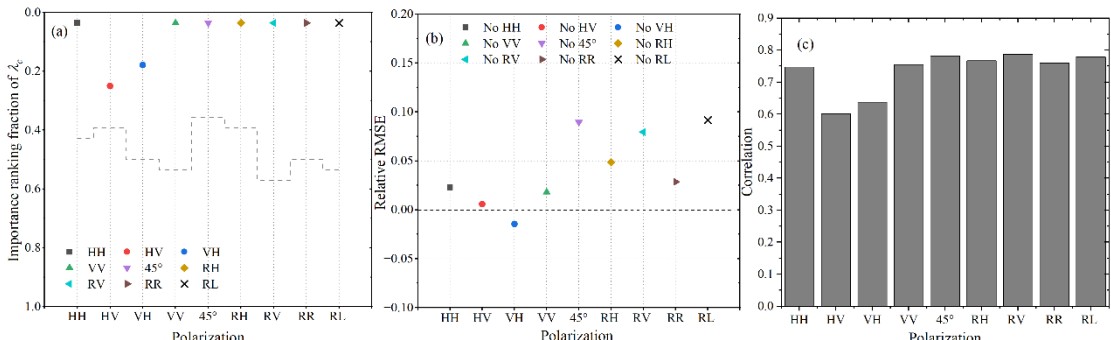

**Figure 20.** The importance of $\lambda_c/\beta$ to SWH retrieval from single-polarization models. (**a**) The importance ranking of $\lambda_c/\beta$ provided by XGBoost models normalized to 0–1. The dotted line represents the optimal feature sets under different polarizations. (**b**) The effect of removing $\lambda_c/\beta$ on the SWH estimation from different polarized models. Different markers in various colors represent $\lambda_c/\beta$ of different polarizations. (**c**) The correlation between $\lambda_c/\beta$ and ERA5 SWH under different polarizations.

The importance of $\lambda_c/\beta$ to SWH retrieval under combined-polarizations is shown in Figure 21. In the HH + HV, VV + VH, and RL + HV models, the co-polarized $\lambda_c/\beta$ ranks high, and the cross-polarized $\lambda_c/\beta$ ranks low. And their $\lambda_c/\beta$-terms are selected by the optimal feature sets except for the HV $\lambda_c/\beta$ in the RL + HV model. However, the performance of these three models is almost unchanged or even significantly improved after removing the $\lambda_c/\beta$-terms. In another dual-polarization model, RH + RV, the $\lambda_c/\beta$-terms rank high. There is a significant positive effect of RH $\lambda_c/\beta$ and a negative effect of RV $\lambda_c/\beta$. Overall, the performance of the model after removing both RH and RV $\lambda_c/\beta$ is still significantly degraded. Only HV $\lambda_c/\beta$ is rejected by the optimal feature set in the Quad model, and both HV- and VH-polarized $\lambda_c/\beta$ are rejected in the All model. The effects of these rejects on the model performance are also confirmed to be small or negative. It is clear that most $\lambda_c/\beta$-terms negatively affect the model accuracy under the combined-polarizations including HV or VH. This may be caused by the nonlinearity between the $\lambda_c/\beta$-terms and the HV or VH features.

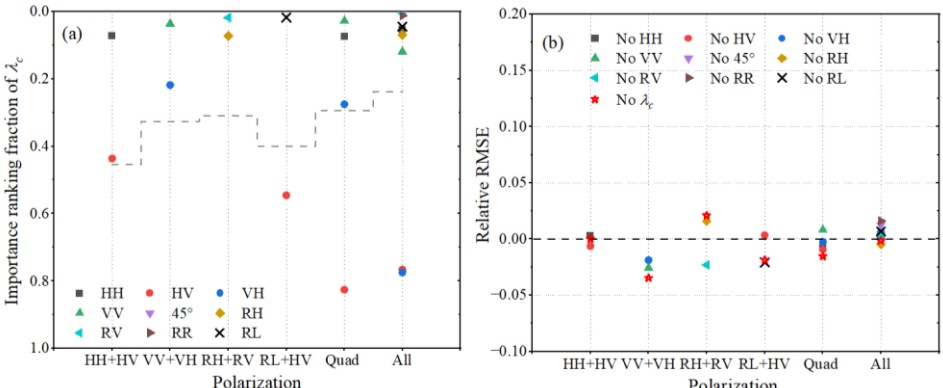

**Figure 21.** The importance of $\lambda_c/\beta$ to SWH retrieval from combined-polarization models. (**a**) The importance ranking of $\lambda_c/\beta$ provided by XGBoost models, normalized to the range of 0–1. (**b**) The effect of removing $\lambda_c/\beta$ on the SWH estimation from different polarized models. The red pentagrams represent the relative RMSE corresponding to the removal of $\lambda_c/\beta$-terms in the combined-polarization models.

### 4.5. The Importance of $\lambda_p$ and $\varphi$

It can be seen from Figures 22–25 that both $\lambda_p$ and $\varphi$ are of low importance to SWH inversion under all the polarizations. Both features are not selected by the optimal feature sets due to their low importance rankings provided by XGBoost. In addition, the two have almost no effect on the accuracy of SWH inversion, which is consistent to their low correlations with ERA5 SWH. We notice that for the VH model, although the ranking of $\lambda_p$ and its correlation with SWH is very low, it still has a little positive effect on the inversion accuracy of SWH, which indicates that the nonlinear effects between features will not only cause negative effects on the model but may also cause positive effects. Therefore, $\lambda_p$ and $\varphi$ are considered to be not very necessary for constructing an empirical model to retrieve SWH.

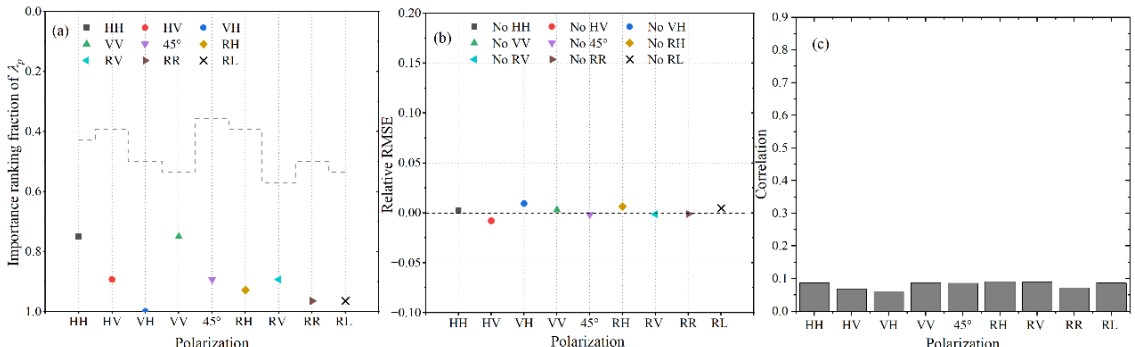

**Figure 22.** The importance of $\lambda_p$ to SWH retrieval from single-polarization models. (**a**) The importance ranking of $\lambda_p$ provided by XGBoost models normalized to 0–1. The dotted line represents the optimal feature sets under different polarizations. (**b**) The effect of removing $\lambda_p$ on the SWH estimation from different polarized models. Different markers in various colors represent $\lambda_p$ of different polarizations. (**c**) The correlation between $\lambda_p$ and ERA5 SWH under different polarizations.

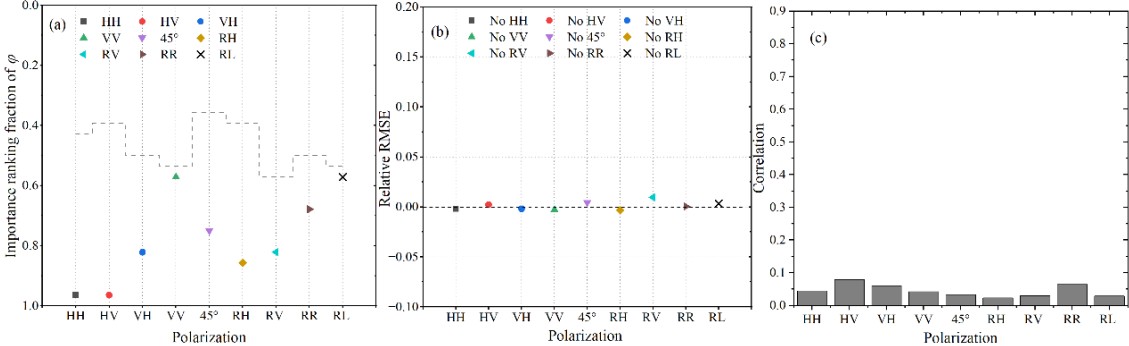

**Figure 23.** The importance of $\varphi$ to SWH retrieval from single-polarization models. (**a**) The importance ranking of $\varphi$ provided by XGBoost models normalized to 0–1. The dotted line represents the optimal feature sets under different polarizations. (**b**) The effect of removing $\varphi$ on the SWH estimation from different polarized models. Different markers in various colors represent $\varphi$ of different polarizations. (**c**) The correlation between $\varphi$ and ERA5 SWH under different polarizations.

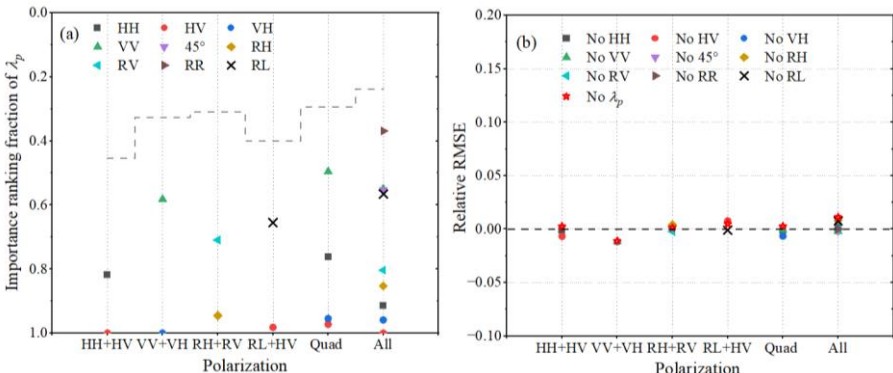

**Figure 24.** The importance of $\lambda_p$ to SWH retrieval from combined-polarization models. (**a**) The importance ranking of $\lambda_p$ provided by XGBoost models, normalized to the range of 0–1. (**b**) The effect of removing $\lambda_p$ on the SWH estimation from different polarized models. The red pentagrams represent the relative RMSE corresponding to the removal of $\lambda_p$-terms in the combined-polarization models.

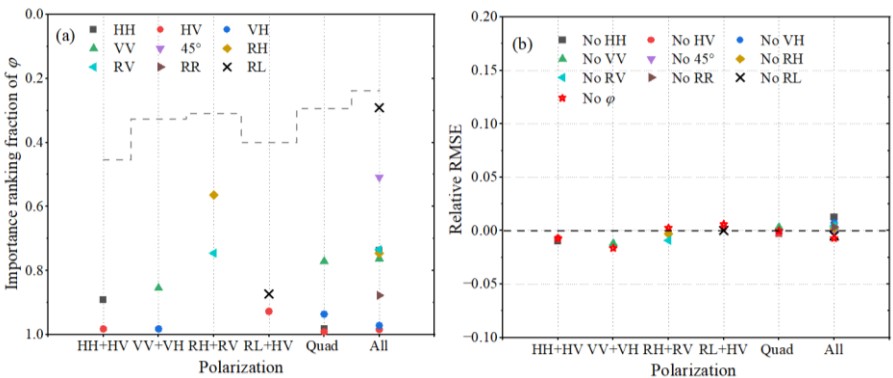

**Figure 25.** The importance of $\varphi$ to SWH retrieval from combined-polarization models. (**a**) The importance ranking of $\varphi$ provided by XGBoost models, normalized to the range of 0–1. (**b**) The effect of removing $\varphi$ on the SWH estimation from different polarized models. The red pentagrams represent the relative RMSE corresponding to the removal of $\varphi$-terms in the combined-polarization models.

### 4.6. The Importance of CWAVE Spectral Parameters

The performance change of each model for SWH retrieval is caused by removing 20 CWAVE spectral parameters is explored, and the relative RMSE is shown in Figure 26. For the single-polarization models, the removal of spectral parameters makes the model performance drop significantly. The influence of spectral parameters on the cross-polarization models is obviously smaller than that of the co-polarization and hybrid-polarization models. For the combined-polarization models, the influence of the cross-polarized spectral parameters on the model performance is relatively small. And it is worth noting that in the VV + VH model, the removal of the VH spectral parameters makes the model perform better. In addition, the performance degradation of the combined-polarization models with all spectral parameters removed is serious. That is to say, there is indeed some additional significant information in the spectral parameters, which is helpful for SWH estimation.

The 20 spectral parameters have different effects on the performance of the models. We take the two spectral parameters $S_1$ and $S_{19}$ (with different rankings) as examples to analyze the importance of a single spectral parameter to the SWH inversion. We can see from Figures 27 and 28 that in the single-polarization models, $S_1$ is more important for SWH inversion than $S_{19}$. The former is selected into the optimal feature set in all single-polarization models, while the latter is not. Besides, $S_1$ also has a more significant impact on model accuracy than $S_{19}$. This is mainly because the correlation of the former with ERA5 SWH is significantly greater than that of the latter. The spectral parameter $S_1$ correlates higher with ERA5 SWH in HV and VH polarizations, but it not only obtains a

relatively low importance ranking but also has little effect on the model accuracy and even has a noticeable negative effect. This negative effect in HV may be caused by the nonlinear interaction between $S_1$ and other features (such as *cvar*, which is already described in the section of *the importance of cvar*). In addition, the $S_1$ shows a slightly larger effect on the SWH inversion in the hybrid-polarizations (45° linearly, RH, and RV) models than in other polarization models. This trend is also reflected by $S_{19}$, but does not always hold for the other spectral parameters (the results are not reported here).

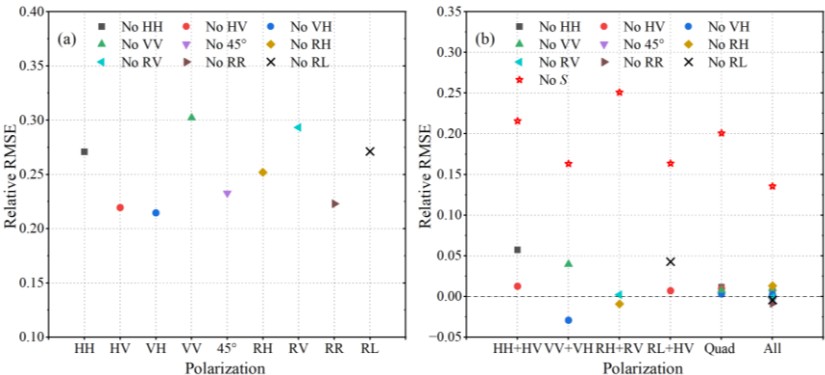

**Figure 26.** The importance of 20 CWAVE spectral parameters to SWH retrieval from (**a**) single-polarization models, and (**b**) combined-polarization models.

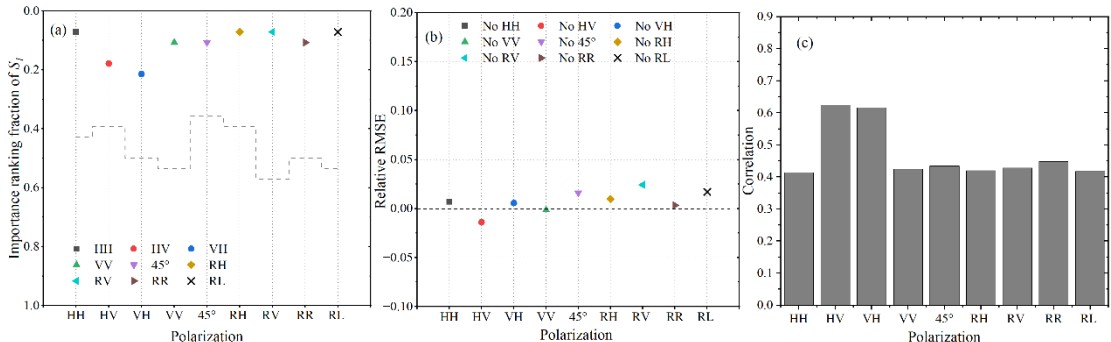

**Figure 27.** The importance of $S_1$ to SWH retrieval from single-polarization models. (**a**) The importance ranking of $S_1$ provided by XGBoost models normalized to 0–1. The dotted line represents the optimal feature sets under different polarizations. (**b**) The effect of removing $S_1$ on the SWH estimation from different polarized models. Different markers in various colors represent $S_1$ of different polarizations. (**c**) The correlation between $S_1$ and ERA5 SWH under different polarizations.

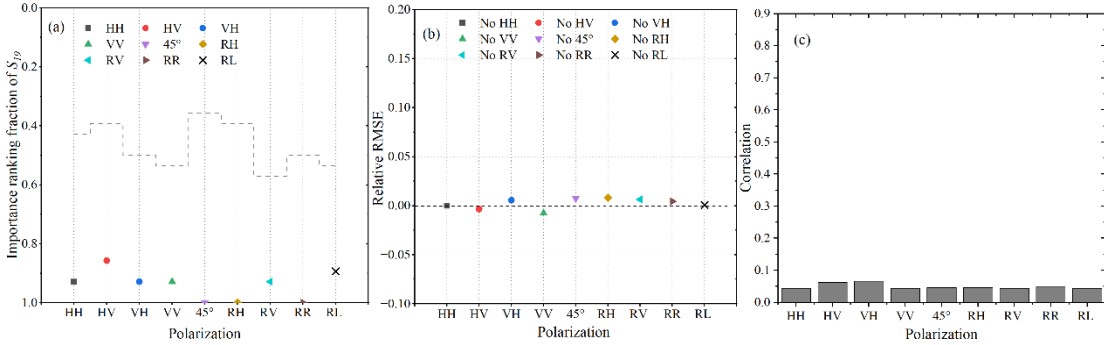

**Figure 28.** The importance of $S_{19}$ to SWH retrieval from single-polarization models. (**a**) The importance ranking of $S_{19}$ provided by XGBoost models normalized to 0–1. The dotted line represents the optimal feature sets under different polarizations. (**b**) The effect of removing $S_{19}$ on the SWH estimation from different polarized models. Different markers in various colors represent $S_{19}$ of different polarizations. (**c**) The correlation between $S_{19}$ and ERA5 SWH under different polarizations.

As shown in Figures 29 and 30, both $S_1$ and $S_{19}$ exhibit different importance for SWH inversion in the combined-polarization models from that in the single-polarization models. Although the importance ranking provided by XGBoost for $S_1$ is much higher than that for $S_{19}$, both features have little effect on the SWH inversion accuracy of the combined-polarization models. In particular, $S_1$ is selected as a member of the optimal feature sets for all four dual-polarization models, but none of its effects on model accuracy is positive. Taking VV + VH as an example, we re-explore the contribution of $S_1$ to the accuracy of the model using the optimal feature set. The relative RMSE of the model using the optimal feature set reaches 0.013 when the $S_1$-term is removed, i.e., the $S_1$-term has a considerable positive impact on the model accuracy after removing redundant features. In addition, in the Quad and All polarization models, only the HH polarized $S_1$ is rejected by optimal feature sets, but all $S_{19}$-terms are rejected. However, the impact of $S_1$ and $S_{19}$ on the model accuracy is similar.

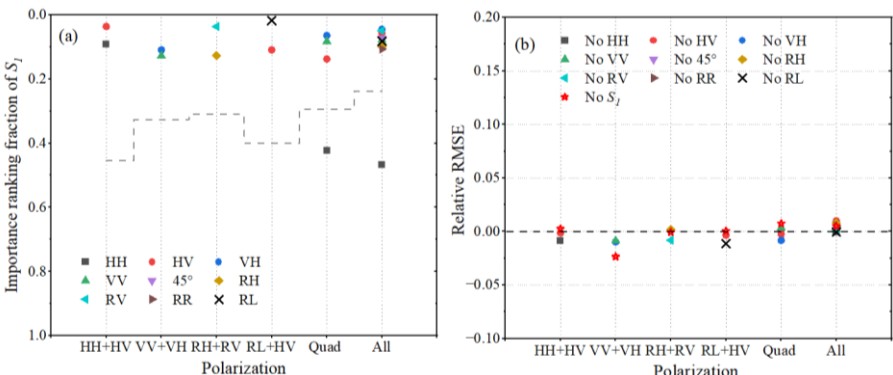

**Figure 29.** The importance of $S_1$ to SWH retrieval from combined-polarization models. (**a**) The importance ranking of $S_1$ provided by XGBoost models, normalized to the range of 0–1. (**b**) The effect of removing $S_1$ on the SWH estimation from different polarized models. The red pentagrams represent the relative RMSE corresponding to the removal of $S_1$-terms in the combined-polarization models.

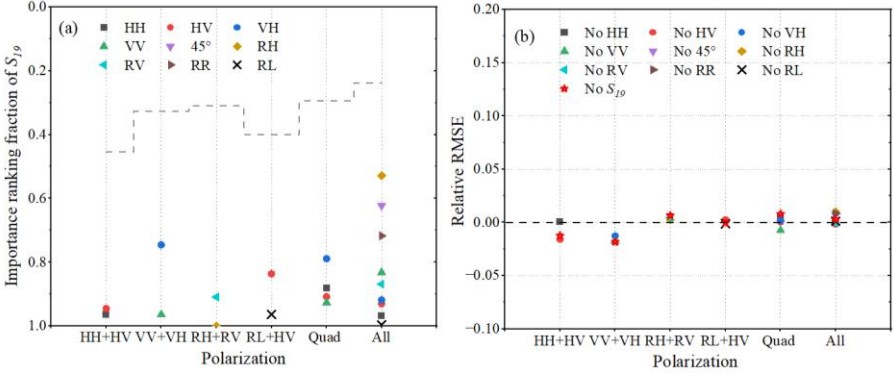

**Figure 30.** The importance of $S_{19}$ to SWH retrieval from combined-polarization models. (**a**) The importance ranking of $S_{19}$ provided by XGBoost models, normalized to the range of 0–1. (**b**) The effect of removing $S_{19}$ on the SWH estimation from different polarized models. The red pentagrams represent the relative RMSE corresponding to the removal of $S_{19}$-terms in the combined-polarization models.

*4.7. The Importance of θ*

As shown in Figure 31, the importance of $\theta$ to the SWH inversion is relatively large under all the polarizations. It can be seen from Figure 31a, the rankings of $\theta$ under different polarizations are similar. It ranks slightly higher in RH polarization and slightly lower in VH polarization. The $\theta$ is selected into the optimal feature sets as an important feature in single- and combined- polarization models due to its high ranking. As shown in Figure 31b, the $\theta$ significantly affects the SWH estimation accuracy under all the polarizations, but

the effects differ between different models. The effect on RR model is the largest in the single-polarization models, and that of RH + RV model is the largest in the combined-polarization models. However, this effect is more pronounced for the single-polarization model, probably because of the fewer input features of the single-polarization models.

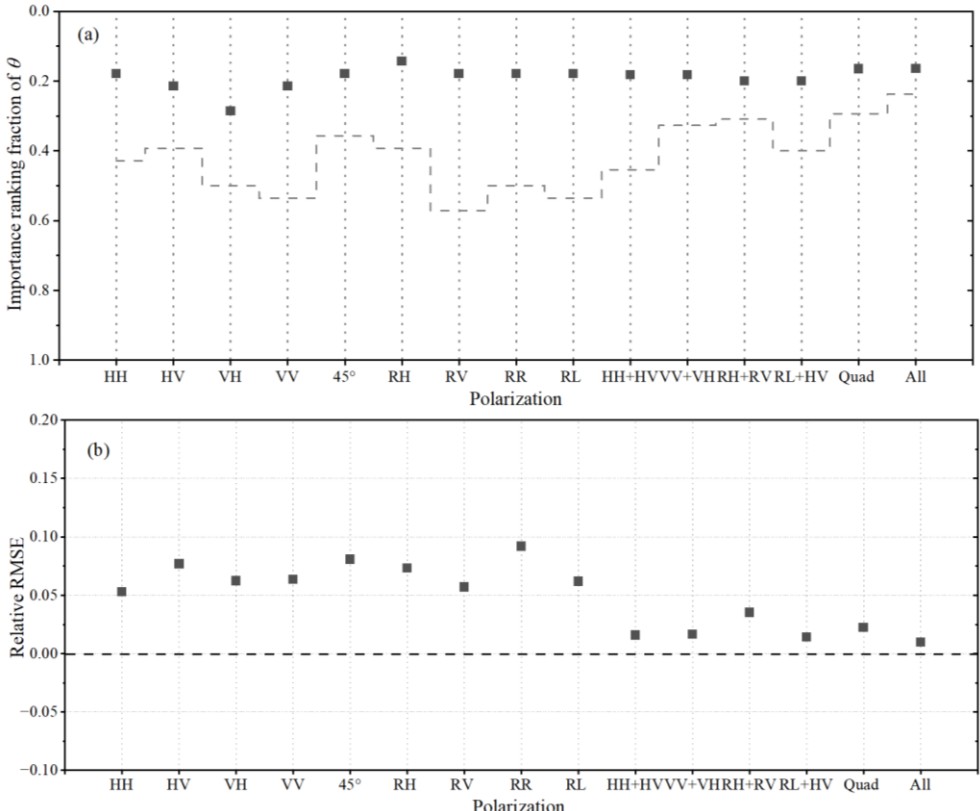

**Figure 31.** The importance of $\theta$ to SWH retrieval from different polarization models. (**a**) The importance ranking of $\theta$ provided by XGBoost models, normalized to the range of 0–1. (**b**) The effect of removing $\theta$ on the SWH estimation from different polarized models.

In summary, the importance of 28 features to the SWH inversion from polarimetric Gaofen-3 wave mode imagettes using XGBoost model is different. The NRCS, $\lambda_c/\beta$, $S_1$, and $\theta$ not only receive a high importance ranking, but also obviously and positively affect the performance of SWH inversion. In contrast, the *cvar*, *skew*, *kurt*, $\lambda_p$, $\varphi$, and $S_{19}$ are of little importance to the model. The low contributions of *cvar* and *skew* mainly result from repeatability or nonlinearity with other features. In addition, for a certain feature, their importance to the SWH inversion model is different under different polarizations. For example, NRCS has a greater impact on SWH inversion under cross-polarizations (HV, VH, and RR), while the cutoff wavelength has a weaker effect under HV and VH polarizations. Furthermore, removing all the 20 CWAVE spectral parameters has smaller effect in cross-polarizations than in other polarizations, whereas this difference is not evident when only a single spectral parameter is removed. The reason for the different importance of 28 SAR features to SWH estimation is not only the correlation between features and SWH but also the complex interaction between features. The negative effect on the model accuracy caused by the interaction between features is more likely to occur in VV + VH polarization.

## 5. Conclusions

The XGBoost models are established based on the collocated data set of approximately 12,000 Gaofen-3 wave mode SAR imagettes matched with the SWH from ERA5 reanalysis to retrieve SWH using features of different polarizations in this study. The optimal feature set is determined for each model as the performance of the model no longer improves

when the fraction of input features exceeds 0.4. Cross-polarization exhibits a non-negligible observational advantage at high sea states. The combined-polarization modes such as HH + HV, VV + VH, and Quad (HH + HV+ VH + VV) and All (HH + HV + VH + VV + 45° + RH + RV + RR + RL) achieve better performance of SWH retrieval than single-polarizations by combining the advantages of co-polarization and cross-polarization. Then the prediction accuracies of the models are independently assessed based on the collocations of SAR-buoy and SAR-altimeter. All the models are confirmed to provide a reliable estimation of SWH. The HH + HV model achieves the best performance with an RMSE of 0.334 m compared to the altimeter, which could be compared with the existing advanced research.

On this basis, the importance of each feature to different polarization models is further discussed. It can be found that the importance of 28 SAR features to the SWH inversion from XGBoost model is not the same. The NRCS, $\lambda_c/\beta$, and $\theta$ have more decisive contributions, while $\lambda_p$ and $\varphi$ have almost no contribution. Moreover, the NRCS has a more substantial effect under cross-polarization, while the $\lambda_c/\beta$ is weaker under HV and VH polarizations. In addition to the correlation between features and SWH, the nonlinearity and repeatability caused by the complex interaction between the features can affect the influence of the features on the SWH estimation accuracy.

**Author Contributions:** Conceptualization, C.F. and Q.Y.; methodology, T.S.; software, T.S.; validation, Y.W., J.M. and J.Z.; writing—original draft preparation, Q.Y.; writing—review and editing, C.F.; funding acquisition, J.Z. All authors have read and agreed to the published version of the manuscript.

**Funding:** This research was funded by National Natural Science Foundation of China (NSFC), grant number 61931025, by the Key Program of Joint Fund of the National Natural Science Foundation of China and Shandong Province under Grant U2006207, and by the Fund of Technology Innovation Center for Ocean Telemetry, Ministry of Natural Resources under Grant 2022003.

**Data Availability Statement:** Not applicable.

**Acknowledgments:** The authors would like to thank Chinese National Satellite Ocean Application Service (NSOAS) for providing the Gaofen-3 SAR wave mode data via the website of https://osdds.nsoas.org.cn/ (accessed on 25 October 2022, registration required). The authors would also like to thank the American National Oceanic and Atmospheric Administration (NOAA) NDBC for providing the buoy data, thank AVISO for providing the Jason-3 altimeter data, and thank ECMWF for providing the ERA5 wind and wave reanalysis data.

**Conflicts of Interest:** The authors declare no conflict of interest.

## Appendix A

**Table A1.** The optimal features used in different polarization models.

| Polarization | Features Selected in Optimal Model (Ranked from Highest to Lowest Importance) |
| --- | --- |
| HH | $\lambda_{c\_HH}$, $S_{1\_HH}$, $S_{6\_HH}$, $S_{16\_HH}$, $\theta$, $\sigma_{0\_HH}$, $S_{10\_HH}$, $S_{8\_HH}$, $S_{11\_HH}$, $S_{13\_HH}$, $S_{18\_HH}$, $S_{2\_HH}$ |
| HV | $S_{3\_HV}$, $S_{6\_HV}$, $S_{10\_HV}$, $\sigma_{0\_HV}$, $S_{1\_HV}$, $\theta$, $\lambda_{c\_HV}$, $skew\_{HV}$, $cvar\_{HV}$, $S_{8\_HV}$, $S_{11\_HV}$ |
| VH | $S_{3\_VH}$, $S_{6\_VH}$, $\sigma_{0\_VH}$, $skew\_{VH}$, $\lambda_{c\_VH}$, $S_{1\_VH}$, $S_{10\_VH}$, $\theta$, $cvar\_{VH}$, $S_{16\_VH}$, $S_{11\_VH}$, $S_{5\_VH}$, $S_{13\_VH}$, $S_{8\_VH}$ |
| VV | $\lambda_{c\_VV}$, $S_{6\_VV}$, $S_{1\_VV}$, $S_{16\_VV}$, $S_{13\_VV}$, $\theta$, $S_{11\_VV}$, $S_{18\_VV}$, $S_{2\_VV}$, $S_{8\_VV}$, $S_{9\_VV}$, $S_{10\_VV}$, $\sigma_{0\_VV}$, $S_{12\_VV}$, $skew\_{VV}$ |

**Table A1.** *Cont.*

| Polarization | Features Selected in Optimal Model (Ranked from Highest to Lowest Importance) |
|---|---|
| 45°linearly | $\lambda_{c\_45}, S_{6\_45}, S_{1\_45}, S_{16\_45}, \theta, S_{11\_45}, \sigma_{0\_45}, skew_{\_45}, S_{2\_45}, S_{18\_45}$ |
| RH | $\lambda_{c\_RH}, S_{1\_RH}, S_{6\_RH}, \theta, S_{16\_RH}, S_{11\_RH}, S_{10\_RH}, \sigma_{0\_RH}, S_{2\_RH}, skew_{\_RH}, S_{18\_RH}$ |
| RV | $\lambda_{c\_RV}, S_{1\_RV}, S_{6\_RV}, S_{16\_RV}, \theta, S_{11\_RV}, S_{18\_RV}, S_{10\_RV}, S_{2\_RV}, S_{8\_RV}, S_{9\_RV},$ $S_{13\_RV}, skew_{\_RV}, S_{12\_RV}, kurt_{\_RV}, \sigma_{0\_RV}$ |
| RR | $\lambda_{c\_RR}, S_{6\_RR}, S_{1\_RR}, \sigma_{0\_RR}, \theta, S_{11\_RR}, S_{13\_RR}, S_{10\_RR}, S_{2\_RR}, S_{16\_RR}, S_{8\_RR},$ $cvar_{\_RR}, S_{9\_RR}, S_{18\_RR}$ |
| RL | $\lambda_{c\_RL}, S_{1\_RL}, S_{6\_RL}, S_{16\_RL}, \theta, S_{11\_RL}, skew_{\_RL}, S_{10\_RL}, S_{2\_RL}, S_{18\_RL}, S_{9\_RL},$ $S_{8\_RL}, S_{13\_RL}, kurt_{\_RL}, \sigma_{0\_RL}$ |
| HH + HV | $S_{3\_HV}, S_{6\_HV}, \sigma_{0\_HV}, \lambda_{c\_HH}, S_{1\_HV}, S_{6\_HH}, cvar_{\_HV}, skew_{\_HV}, S_{16\_HH}, \theta, S_{10\_HV},$ $S_{1\_HH}, skew_{\_HH}, S_{10\_HH}, S_{16\_HV}, kurt_{\_HH}, S_{8\_HV}, S_{11\_HH}, S_{11\_HV}, S_{8\_HH}, S_{9\_HV},$ $S_{4\_HH}, cvar_{\_HH}, \lambda_{c\_HV}, S_{2\_HH}$ |
| VV + VH | $S_{3\_VH}, \lambda_{c\_VV}, S_{6\_VH}, \sigma_{0\_VH}, S_{6\_VV}, S_{1\_VH}, S_{1\_VV}, S_{16\_VV}, skew_{\_VH}, \theta, S_{11\_VV},$ $\lambda_{c\_VH}, S_{4\_VV}, cvar_{\_VH}, skew_{\_VH}, S_{10\_VV}, kurt_{\_VV}, S_{10\_VH}$ |
| RH + RV | $\lambda_{c\_RV}, S_{1\_RV}, S_{16\_RH}, \lambda_{c\_RH}, S_{6\_RV}, S_{6\_RH}, S_{1\_RH}, S_{16\_RV}, S_{18\_RV}, S_{10\_RV}, \theta,$ $S_{2\_RV}, S_{9\_RV}, \sigma_{0\_RH}, skew_{\_RV}, S_{11\_RV}, S_{8\_RV}$ |
| RL + HV | $\lambda_{c\_RL}, S_{6\_HV}, skew_{\_HV}, S_{3\_HV}, \sigma_{0\_HV}, S_{1\_HV}, S_{6\_RL}, S_{16\_RL}, cvar_{\_HV}, S_{1\_RL}, \theta,$ $kurt_{\_HV}, S_{16\_HV}, skew_{\_RL}, S_{11\_HV}, S_{11\_RL}, S_{13\_HV}, S_{10\_HV}, kurt_{\_RL}, S_{8\_HV}$ |
| Quad | $S_{3\_HV}, \sigma_{0\_VH}, \lambda_{c\_VV}, S_{6\_VH}, S_{3\_VH}, S_{6\_HV}, S_{1\_VH}, \lambda_{c\_HH}, S_{1\_VV}, cvar_{\_VH}, S_{6\_VV},$ $S_{16\_VV}, cvar_{\_HV}, \sigma_{0\_HV}, S_{1\_HV}, S_{16\_HH}, skew_{\_HV}, \theta, skew_{\_VH}, S_{6\_HH}, S_{10\_HV},$ $cvar_{\_VV}, S_{11\_VV}, S_{16\_HV}, S_{8\_VV}, S_{10\_HH}, skew_{\_VV}, kurt_{\_VV}, \sigma_{0\_VV}, \lambda_{c\_VH},$ $S_{13\_HV}, S_{7\_VV}$ |
| All | $\lambda_{c\_RV}, \lambda_{c\_45}, \lambda_{c\_RR}, S_{6\_HV}, S_{6\_VH}, \sigma_{0\_VH}, S_{6\_RH}, S_{16\_RV}, skew_{\_HV}, \lambda_{c\_RL}, S_{1\_VH},$ $S_{1\_RV}, \lambda_{c\_HH}, S_{1\_HV}, \sigma_{0\_HV}, S_{16\_RL}, \lambda_{c\_RH}, S_{1\_45}, S_{1\_VV}, S_{1\_RL}, skew_{\_VH},$ $cvar_{\_HV}, cvar_{\_VH}, S_{1\_RH}, S_{16\_RH}, S_{1\_RR}, S_{14\_RL}, S_{11\_RV}, \lambda_{c\_VV}, S_{6\_HH}, \sigma_{0\_RR},$ $S_{16\_HH}, S_{2\_RH}, kurt_{\_45}, S_{13\_RH}, S_{6\_VV}, cvar_{\_RV}, S_{6\_RL}, cvar_{\_RR}, \theta, S_{13\_VV}, S_{11\_VV},$ $S_{11\_HV}, S_{13\_RV}, skew_{\_RV}, S_{11\_45}, cvar_{\_RL}, S_{8\_RR}, cvar_{\_45}, S_{18\_RR}, S_{16\_45}, S_{6\_45},$ $S_{2\_VV}, skew_{\_VV}, S_{18\_VV}, S_{11\_RL}, skew_{\_RR}, S_{4\_HH}, S_{18\_RV}, S_{8\_RV}, S_{11\_RR}, \sigma_{0\_RL},$ $S_{8\_RL}, S_{4\_RL}, S_{3\_RL}, S_{4\_RR}, S_{9\_RL}, S_{10\_RH}, cvar_{\_VV}, S_{9\_RV}$ |

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
