# Peer review of "Significant Wave Height Retrieval Using XGBoost from Polarimetric Gaofen-3 SAR and Feature Importance Analysis"

_remotesensing, doi:10.3390/rs15010149_

Round 1
Reviewer 1 Report
To my opinion, the paper is timely and shall be well received within the interested community.
I would invite the authors to provide a clear definition of the S_n parameters, especially those commented in the paper. While mentioned and discussed in the abstract and in the paper, S1 or S15, for instance, are not defined. To ease the presentation, I would invite the authors to use one particular case, the definition of the Sn parameters, and the results applied to the given image.
I would also invite the authors to discuss the impact of winds on the cross-pol measurements. At low wind, VH, HV products will be more impacted by noise (the signal is too weak). Is this identified from the present analysis ?
Reviewer 2 Report
It is an interesting topic to retrieve significant wave height from Gaofen-3 SAR wave mode data and to analyze the importance of various polarized features for SAR SWH retrieval. The XGBoost model provides the possibility for this research. Therefore, in this paper, the XGBoost models are developed for SWH retrieval, and the importance of each SAR feature for SWH inversion is analyzed in different polarization modes. This study is meaningful, but still needs further improvement in the following aspects:
Major comments:
1. The descriptions of polarization data, polarization features, and polarization models are confusing, vague and ambiguous. It is recommended to unify and sort out the relevant descriptions in the full text.
2. One of the main research contents of this paper is the importance analysis of features. However, the necessity of carrying out this study needs to be further explained and proved.
3. For the 20 CWAVE orthogonal spectral parameters extracted from Gaofen-3 SAR, only two spectral features are discussed in the analysis of feature importance, while other spectral features are ignored. Is this reasonable? Furthermore, why are the two spectral parameters S1 and S19 chosen?
4. English presentation needs to be improved, and some sentences should be rearranged for clarity, and some grammatical problems need to be further checked.
Minor comments:
1. On page 2, line 56, the description of the reason why the machine learning model has become the mainstream method for SAR SWH inversion estimation is not accurate.
2. On page 7, lines 236-237, the sentence “Many studies have been carried out to explore the empirical relationship of λc with wind speeds and sea states” is unnecessary, it is recommended to delete.
3. On page 8, lines 316-317, the descriptions of the performance of the two polarization models (HH+HV and VV+VH) are not accurate, actually, only VV+VH model use less features than single-polarizations with the best feature set. Besides, the reason for this is worthy of further exploration.
Reviewer 3 Report
The paper "Significant wave height retrieval using XGBoost from polarimetric Gaofen-3 SAR and feature importance analysis" is dedicated to the XGBoost model's performance in applying to Gaofen-3 SAR images. The paper examines model performance for different polarization-mode using buoy and altimeter collocations. The detailed analysis provided for multiple model feature performance regarding the efficiency of SWH inversion.
While the subject of the paper is very technical, rather scientific, the paper contains very detailed analysis of XGBoost performance for SAR SWH retrieval in all possible modes. As well, the role of different optimal features is examined in details for all possible inversion modes.
I believe the information in the paper will be useful for the specialists working in radar ocean remote sensing domain.
Round 2
Reviewer 2 Report
This paper has been revised carefully. Some errors are corrected. Some analyses are added.